# RNF114 and RNF166 exemplify reader-writer E3 ligases that extend K11 polyubiquitin onto sites of MARUbylation

Rachel E Lacoursiere [ID] [1,4], Kapil Upadhyaya [ID] [2,4], Jasleen Kaur Sidhu [ID] [1], Ivan Rodriguez Siordia [2], Daniel S Bejan [ID] [2], Michael S Cohen [ID] [2,3] ✉ & Jonathan N Pruneda [ID] [1,2,3] ✉

## Abstract

**Ubiquitin (Ub) cooperates with other post-translational modifications to provide a tiered opportunity for protein regulation. Deltex E3 ligases were previously implicated in ubiquitylation of ADP-ribose (ADPr)-containing macromolecules in vitro, generating a noncanonical mono-ADPr-Ub ester (MARUbe). We previously identified mono-ADPr ubiquitylation (MARUbylation) on PARP7 in cells, which was extended with K11-linked polyUb, suggesting an intricately regulated, multilayered post-translational modification. Here, we show that the Deltex DTX2 ubiquitylates ADPr modifications on PARP7 in cells, which depends on PARP7 catalytic activity. We further identify RNF114 as the E3 ligase responsible for K11-linked polyUb extension on sites of PARP7 MARUbylation. Using a chemoenzymatic approach, we developed a fluorescent Ub-ADPr probe and find that RNF114 explicitly recognizes MAR-Ubylated species. We used AlphaFold3 to examine the mechanisms of Ub-ADPr recognition and K11-linked polyUb extension by RNF114. We identify a tandem Di19-UIM module in RNF114 as a MARUbe-binding domain (M-UBD), thus providing a reader function that interfaces with K11-specific writer activity. Finally, we describe a small family of M-UBD-containing E3 ligases that demonstrate preference for Ub-ADPr, which we call MARUbe-Targeted Ligases (M-UTLs).**

**Keywords** ADP-ribose Ubiquitylation; K11 Polyubiquitylation; Post-translational Modifications; PARP/Ubiquitin Crosstalk
**Subject Categories** Post-translational Modifications & Proteolysis; Structural Biology

See also: AJ Middleton & CL Day

## Introduction

Ubiquitylation is one of the most prevalent and versatile post-translational modifications (PTMs) in the eukaryotic cell. Concerted efforts from three classes of ubiquitylating enzymes (E1 activating enzymes, E2 conjugating enzymes, and E3 ligases) facilitate the covalent modification of a substrate with the small protein ubiquitin (Ub). Canonically, ubiquitylation occurs at lysine sidechains in a proteinaceous substrate, and repeated cycles of the cascade can result in the formation of polyubiquitin (polyUb) chains through Ub-Ub linkages (Swatek and Komander, 2016).

As a protein itself, Ub is susceptible to regulation by other PTMs such as phosphorylation and acetylation. For example, phosphorylation of Ub at Ser65 by the kinase PINK1 acts as a crucial initial signal for Parkin/PINK1-dependent mitophagy (Kane et al, 2014; Kazlauskaite et al, 2014; Koyano et al, 2014). Acetylation of Ub lysine residues alters the efficiency of E2~Ub conjugate formation (Lacoursiere and Shaw, 2021). Further, Ub signaling can be modulated through the activity of bacterial effectors that target Ub for ADP-ribosylation at various sidechains (Qiu et al, 2016; Yan et al, 2020). Human enzymes, including PARPs and the Deltex family of E3 ligases, have also been reported to ADP-ribosylate Ub, though this modification occurs at the C-terminus (Ashok et al, 2022; Chatrin et al, 2020; Yang et al, 2017). While the exact role of this modification is unclear, interplay between ADP-ribosylation and ubiquitylation is prevalent in pathways such as the DNA damage response (DDR) and innate immune signaling (Pellegrino and Altmeyer, 2016).

Crosstalk between ADP-ribosylation and ubiquitylation can be exemplified by poly-ADP-ribosylation (PARylation)-dependent ubiquitylation (PARdU). RNF146, which complexes with the PARylating PARPs Tankyrase1/2 (TNKS1/2; formerly PARP5a/b), facilitates the canonical ubiquitylation of a substrate in a PARylation-dependent manner (Callow et al, 2011; Zhang et al, 2011). PAR recognition occurs at the internal repeating unit *iso*-ADPr by the Trp-Trp-Glu (WWE) PAR "reader" domain of RNF146, which induces a conformational change and places the catalytic E3 ligase RING domain in an active orientation to facilitate catalysis (DaRosa et al, 2015). PARdU demonstrates the cooperation between PARPs and E3 ligases in an ADP-ribosylation-dependent manner. Beyond RNF146, other E3 ligases

[1]Department of Molecular Microbiology and Immunology, Oregon Health & Science University, Portland, OR 97239, USA. [2]Department of Chemical Physiology and Biochemistry, Oregon Health & Science University, Portland, OR 97239, USA. [3]Knight Cancer Institute, Oregon Health & Science University, Portland, OR 97239, USA. [4]These authors contributed equally: Rachel E Lacoursiere, Kapil Upadhyaya. ✉E-mail: cohenmic@ohsu.edu; pruneda@ohsu.edu

contain WWE domains, including the Deltex RING E3 ligases DTX1, DTX2, and DTX4 (Chatrin et al, 2020). Deltex E3 ligases are also characterized by a Deltex carboxy-terminal (DTC) domain at their C-termini. Through in vitro experiments with select family members, this domain has been shown to bind and ubiquitylate $NAD^+$, free ADPr, or even mono-ADP-ribosylated (MARylated) substrates such as proteins and nucleic acids (Ahmed et al, 2020; Chatrin et al, 2020; Zhu et al, 2024; Zhu et al, 2022).

We recently discovered that ubiquitylation of MAR protein modifications occurs in cells as a dual PTM downstream of multiple PARPs under conditions including interferon stimulation (Bejan et al, 2025). We termed this process MAR ubiquitylation (MARUbylation) to distinguish it from MARylation and canonical ubiquitylation. The first step in MARUbylation requires PARP catalytic activity to attach mono-ADP-ribose to a protein substrate, like itself. Next, an E3 Ub ligase recognizing ADPr and/or the MARylated substrate ubiquitylates the MAR through a labile ester linkage, forming the MAR-Ub ester (MARUbe). Although Deltex E3 ligases have not been formally shown to generate the initial MARUbe in cells, an abundance of in vitro experiments supports this hypothesis. Interestingly, we identified K11 polyUb linkages extending from the initial MARUbe, suggesting the activity of an additional K11-specific ligase. Together, MARUbylation represents three layers of PTM regulated by at least as many enzymes.

Multiple lines of evidence connect PARPs with Deltex E3 ligases, providing confidence in a functional role for MARUbylation in cells (Huttlin et al, 2021; Oughtred et al, 2021; Schweppe et al, 2018). The same lines of evidence also identify RNF114 and/or RNF166, two RING-type E3 ligases with related sequence and domain topologies. Previous work has shown that RNF114 responds to ADP-ribose signaling. For example, RNF114 is recruited to sites of DNA damage in an ADP-ribosylation-dependent manner (Djerir et al, 2024; Li et al, 2023). Furthermore, outside of the RING domain, both RNF114 and RNF166 encode $Zn^{2+}$-coordinating drought-induced 19 (Di19) domains that have been shown to interact with ADPr (Longarini et al, 2023), and Ub-interacting motifs that recognize Ub (Giannini et al, 2008). Considering this, we hypothesized that RNF114 may read MARUbe modifications downstream of PARP and Deltex activity using its tandem Di19-UIM module, and using the catalytic RING domain, append K11 polyUb as a third layer of PTM regulation. Indeed, in cellular knockdown experiments, we find that PARP, Deltex, and RNF114 activity are all required for K11-extended MARUbylation. Using a newly generated fluorescent Ub-ADPr substrate, we validated specific recognition and K11 polyUb extension by RNF114. To reveal the mechanism of RNF114 reader/writer function, we used AlphaFold3 to model multi-protein complexes and rationalized RNF114 mutations that validate these functions. Lastly, we found that RNF114 and RNF166 are part of a small family of tandem Di19-UIM module-containing E3 ligases. We term this module the MARUbe-binding domain (M-UBD) and show that E3 ligases possessing this domain can act as MARUbe-targeted ligases (M-UTLs).

# Results

## DTX2 and RNF114 regulate the formation and extension of MARUbe on PARP7 in cells

Our recent study revealed the presence of a unique dual PTM, MARUbe, on PARP7 and PARP10 in cells (Bejan et al, 2025). We also found that MARUbe was extended with K11-linked polyUb.

The main questions that remain are which E3 ligases generate MARUbe and which ones elongate MARUbe with K11-linked polyUb within cells. Addressing this knowledge gap is critical for elucidating the cellular function of MARUbylation. We therefore took a candidate approach, focusing on PARP7 MARUbylation because of the good availability of data on interactors and targets. Moreover, PARP7 is a highly sought-after therapeutic target in cancer (Gozgit et al, 2021) and stroke (Cai et al, 2024), and deciphering the function of PARP7-mediated MARUbylation is critical for understanding its cellular function in these diseases. To identify potential E3 ligases that regulate MARUbylation, we analyzed mass spectrometry (MS) data from three sources: (1) BioID proximity labeling (Rodriguez et al, 2021), (2) Co-IP using FLAG-PARP7 (Zhang et al, 2020), and (3) An analog sensitive chemical genetics (ASCG) approach for identifying direct PARP7 MARylation targets (preprint: Rodriguez Siordia et al, 2025). Six E3 ligases were identified in all three MS datasets (Fig. 1A). DTX2 contains a MAR-binding domain, and was shown by us and others to attach Ub to the 3'OH of the adenosine ribose on a wide array of ADPr-containing species in vitro (Ahmed et al, 2020; Bejan et al, 2025; Dearlove et al, 2024; Kelly et al, 2024; Zhu et al, 2024; Zhu et al, 2022). Likewise, recent studies have shown that RNF114 also contains a MAR-binding domain (Longarini et al, 2023), and BioGRID 4.4 and BioPlex interactome analyses showed that RNF114, and the closely related RNF166, are top interactors of several MARylating PARPs, namely PARP7, PARP10, PARP11, and PARP14 (Huttlin et al, 2021; Oughtred et al, 2021; Schweppe et al, 2018). We therefore hypothesized that DTX2 catalyzes the initial MARUbe mark, and that RNF114 and/or RNF166 were responsible for the subsequent extension of K11 polyUb.

To determine whether DTX2 and RNF114/RNF166 are involved in the formation and extension of MARUbe on PARP7 in HEK 293T cells, we used an siRNA-mediated approach and our IP MARUbylation assay with GFP-PARP7, which we previously used to demonstrate K11-linked polyUb extension of PARP7 MARUbylation in cells (Bejan et al, 2025). Briefly, the IP MARUbylation assay involves transiently expressing tagged Ub (HA-Ub) together with GFP-tagged PARP7 in HEK 293T cells. The GFP-tagged PARP7 proteins are immunoprecipitated using GFP-trap beads. Stringent washing of beads is performed under denaturing conditions (7 M urea and 1% SDS) to remove noncovalently bound Ub. We then treated beads with the Ub esterase TssM* to separate the K11-extended MARUbe species from canonical Ub chains on PARP7 into the supernatant, which can be detected by western blotting by probing for the HA tag (Fig. 1B). We first confirmed that PARP7 MARUbylation was indeed dependent on PARP7 activity by treating transfected cells with the PARP7 inhibitor RBN2397 (Figs. 1C,D and EV1) (Gozgit et al, 2021). RBN2397 eliminated MARylation (Fig. 1C, lanes 1 vs. 3) as well as impaired the K11-extended MARUbe species (lanes 2 vs. 4), demonstrating that PARP7 MARUbylation is largely dependent on PARP7 catalytic activity. Knockdown of DTX2 reduced both the MARUbe itself (~10 kDa; monoUb-ADPr) and the K11-extended MARUbe species (15–20 kDa; polyUb-ADPr) (lanes 6 vs. 10), indicating that DTX2 regulates the initial ubiquitylation of MAR on PARP7 (Fig. 1C,D). This is most apparent after normalizing for the increased levels of PARP7 observed following DTX2 knockdown (Fig. 1D), which suggests that MARUbylation may alter the stability and/or compartmentalization of PARP7. Upon siRNA knockdown of RNF114, we observed a reduction in the K11-extended MARUbe species, but an increase in MARUbe

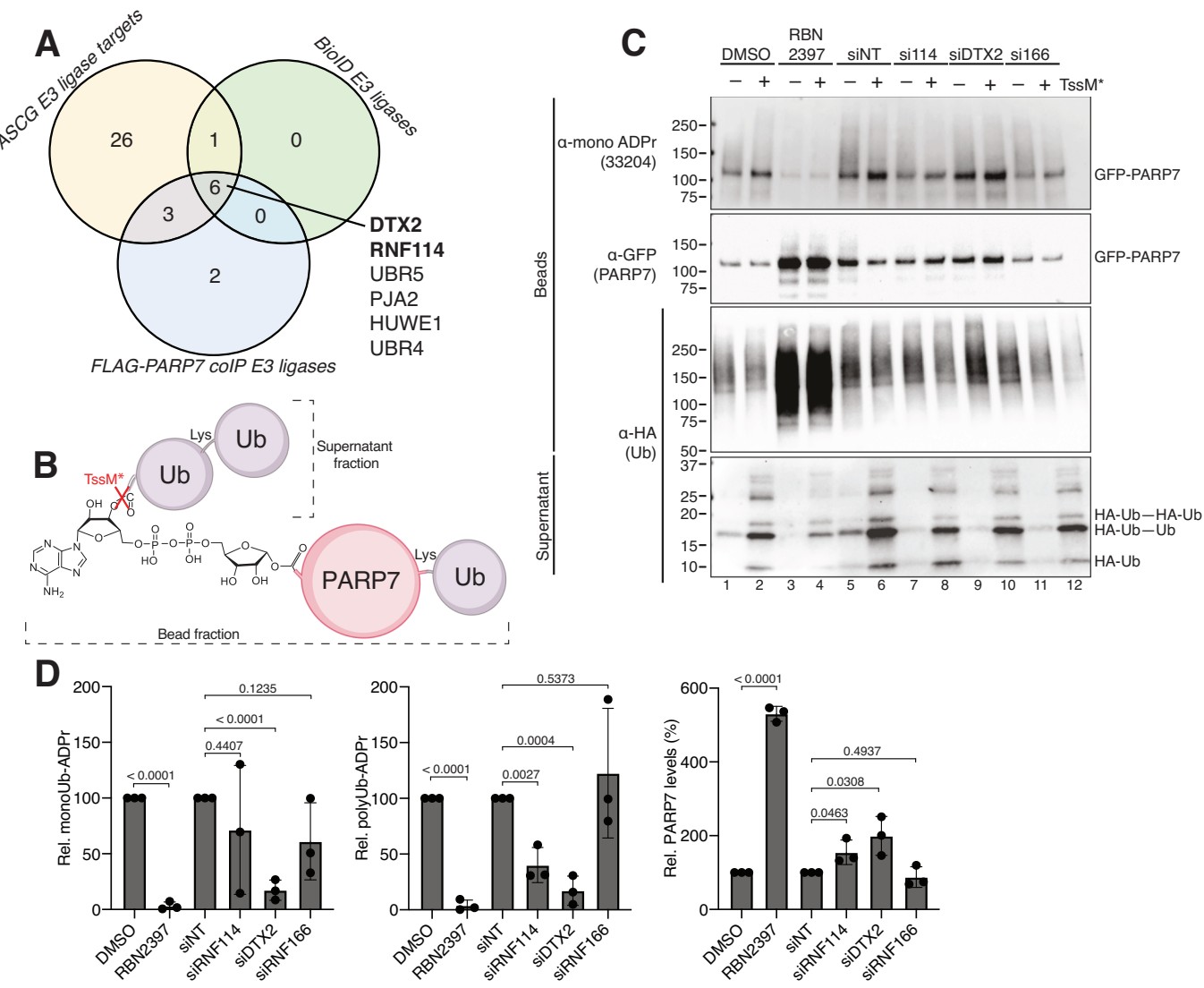

**Figure 1. RNF114 extends PARP7 MARUbylation.**

(A) Identification of six candidate E3 ligases common to three approaches to identify ligases that interact with PARP7. (B) Schematic representation of an IP MARUbylation assay using TssM* to cleave the MAR-bound Ub. In this experiment, GFP-PARP7 is pulled down from cell lysates and subjected to TssM* treatment to release all ubiquitin esters, including the MAR-conjugated ubiquitin. (C) GFP-PARP7 MARUbylated species were pulled down from HEK 293T cells following inhibitor or siRNA treatment as indicated and subjected to a MARUbylation assay using TssM* to separate the stable PARP7 canonical ubiquitylation from the labile MARUbylation. DMSO or NT (non-targeting siRNA) were used as controls for the PARP7 inhibitor RBN2397 treatment and siRNA knockdowns, respectively. Blots shown here are representative of n = 3 biological replicates. (D) Quantification of the released monoUb-ADPr, polyUb-ADPr, and PARP7 levels showing the mean ± standard deviation from n = 3 biological replicates. Band intensities were measured for each experiment and internally referenced to the amount of tubulin present in each sample. Relative protein quantities were assessed such that RBN2397 treatment was referenced to DMSO, and the siE3 ligases were referenced to the siNT. Significance was assessed using an unpaired t test to compare the means of the indicated groups and P values are labeled: ns $P > 0.05$, *$P < 0.05$, **$0.01 > P > 0.001$, ***$0.001 > P > 0.0001$, ****$P < 0.0001$. Source data are available online for this figure.

(Fig. 1C,D, lanes 6 vs. 8). Again, as observed for DTX2, knockdown of RNF114 resulted in increased levels of PARP7, suggesting that specifically K11-linked MARUbylation has a functional impact on PARP7. Surprisingly, following knockdown of RNF166 we were unable to detect any changes in PARP7 levels or MARUbylation status (Fig. 1C,D, lanes 6 vs. 12). The partial reduction in the signal for monoUb-ADPr and polyUb-ADPr observed with all tested siRNA treatments could be due to either incomplete siRNA knockdown or functional redundancy among Deltexes and other E3 ligases (Fig. EV1).

Together, these results support our hypothesis that DTX2 is the major PARP7 MARUbe-generating E3 ligase and RNF114 is the major E3 ligase that extends MARUbe with K11 polyUb.

## An integrated click chemistry and enzymatic strategy yields a novel fluorescent Ub-ADPr tool

Having established RNF114 as a regulator of PARP7 MARUbylation in cells, we next aimed to characterize its in vitro activity to

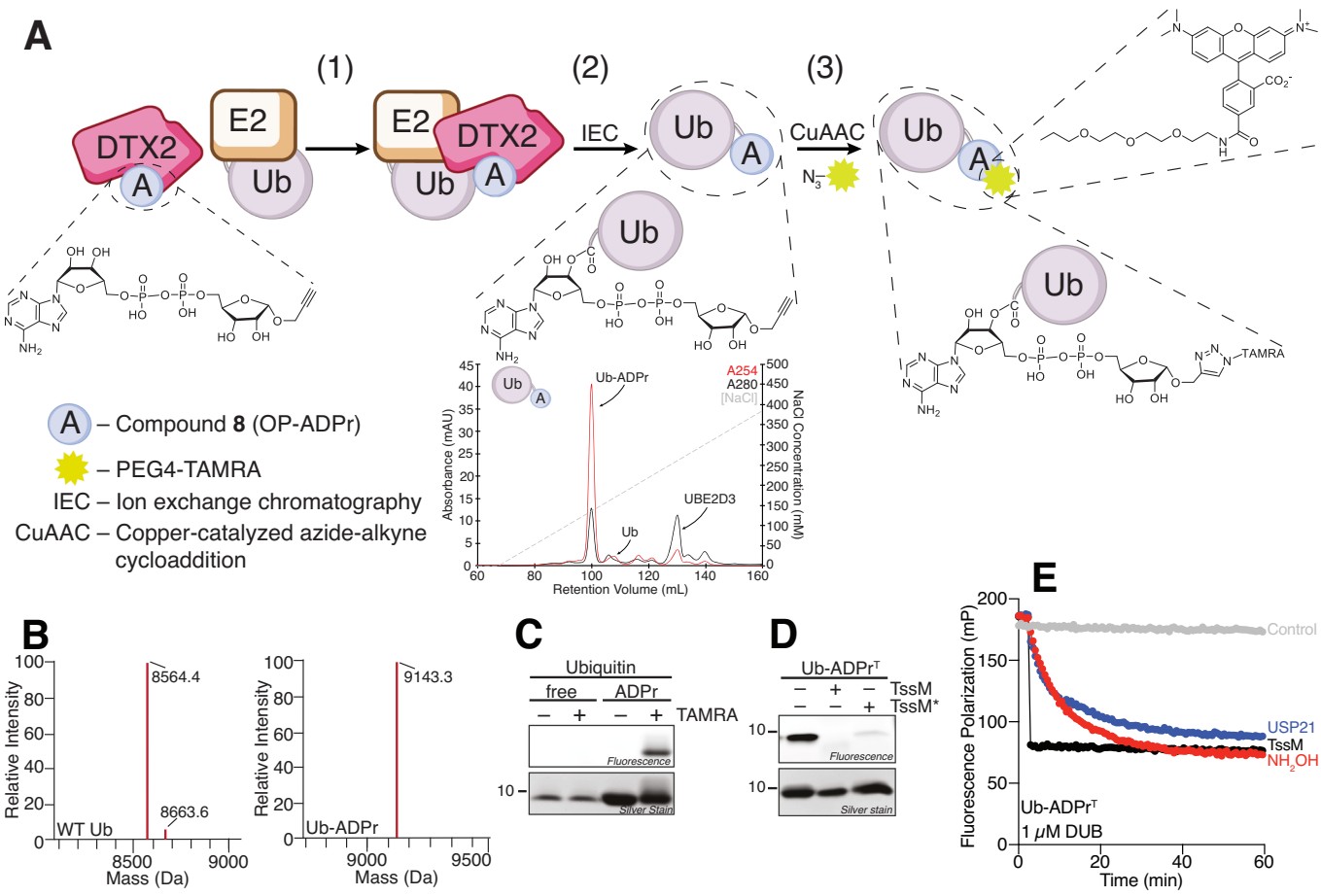

**Figure 2. Synthesis of fluorescent Ub-ADPr (Ub-ADPrᵀ) and ADPrᵀ species.**

(**A**) Schematic representation of the workflow used to produce the fluorescent molecule Ub-ADPrᵀ. The chromatogram provides a representative example for the isolation of Ub-ADPr from other reaction components by ion exchange chromatography. (**B**) Validation of Ub-ADPr by intact mass spectrometry. WT Ub MW$_{expected}$ = 8564.84 Da (left), Ub-ADPr MW$_{expected}$ = 9126.9 Da (right). (**C**) Validation of Ub-ADPrᵀ using in-gel fluorescence before and after the click reaction. (**D**) In-gel fluorescence analysis of a separate experiment treating Ub-ADPrᵀ with TssM or the Ub esterase TssM*. (**E**) Validation of Ub-ADPrᵀ through fluorescence polarization experiments with the indicated deubiquitylase and chemical treatments. Source data are available online for this figure.

understand how RNF114 catalyzes K11-linked ubiquitylation of MARUbe. We envisioned that an ester-linked, fluorescent Ub-ADPr conjugate could serve as a valuable probe for fluorescence polarization (FP)-based assays, enabling real-time monitoring of RNF114 activity, similar to Ub-Lys-TAMRA (Ub-Lysᵀ) for studying canonical ubiquitylation (Geurink et al, 2012). We developed a chemoenzymatic approach to produce a fluorescent ester-linked Ub-ADPr conjugate (Fig. 2A). We first synthesized a clickable ADPr analog containing an α-*O*-propargyl at the anomeric carbon of the ribose ring (compound **8**, OP-ADPr) as described previously, but with minor modifications (Supporting Information) (Liu et al, 2018). We next coupled Ub to OP-ADPr using DTX2 to generate an ester-linked Ub-OP-ADPr (Fig. 2A, step 1) ("Methods"). Ub-OP-ADPr was purified using ion exchange chromatography while monitoring the absorbance at 280 nm (protein) and 254 nm (ADPr). Consistent among many preparations, Ub-OP-ADPr eluted separately from excess Ub and other reaction components on a Resource S column (Fig. 2A, step 2, chromatogram inset). Samples of Ub before and after conjugation to OP-ADPr were

evaluated by intact mass spectrometry (Fig. 2B). The observed mass for Ub (8564.4 Da) was in good agreement with the theoretical mass (8564.8 Da). After conjugation to OP-ADPr, the observed mass was 9143.3 Da, in agreement with the expected mass of 9126.9 Da. We attribute the extra ~16 Da of mass to methionine oxidation of Ub over time. To generate the fluorescent Ub-ADPr analog (Ub-ADPrᵀ), Ub-OP-ADPr was coupled to TAMRA-azide using Copper-Catalyzed Azide-Alkyne Cycloaddition (CuAAC, "click chemistry") (Fig. 2A, step 3). The Ub-ADPrᵀ species was resolved from any unreacted TAMRA-azide using size exclusion chromatography, and SDS-PAGE confirmed the generation of the Ub-ADPrᵀ with in-gel fluorescence (Fig. 2C).

The final Ub-ADPrᵀ product was validated by SDS-PAGE after treatment with the deubiquitylase TssM or the Ub esterase TssM*, which confirmed the expected loss in fluorescence (Fig. 2D). We further validated Ub-ADPrᵀ in a fluorescence polarization experiment using hydroxylamine and the nonspecific deubiquitylases TssM and USP21 to confirm that (1) the Ub is attached to ADPr via an ester linkage; (2) there were no nonspecific interactions between

the Ub and TAMRA-azide, and that the fluorophore was on the ADPr side of Ub-ADPr; and (3) that fluorescence polarization was a valid approach to monitor the state of the Ub-ADPr$^T$ substrate (Fig. 2E). Together, these experiments not only validate the identity of Ub-ADPr$^T$, but also that Ub-ADPr$^T$ is a useful FP probe.

In parallel, we chemically synthesized ADPr$^T$ as described in the Supporting Information and validated it against the known ADPr-binding protein AF1521 (Karras et al, 2005). Using the ADPr$^T$ substrate in a fluorescence polarization assay, we determined the $K_d$ of its interaction with AF1521 to be 1.2 μM (Fig. EV2), which is in good agreement with the previously reported $K_d$ of 3 μM (Nowak et al, 2020). We also showed that while AF1521 had modest affinity for our Ub-containing FP substrates, a clear preference for an ADPr-containing substrate was evident (Fig. EV2). These findings confirm that the TAMRA fluorophore does not interfere with the activity of deubiquitylases or canonical ADPr-binding proteins, so its positioning in Ub-ADPr$^T$ would not be expected to interfere with other protein-protein interactions.

## RNF114 binds ADPr and Ub

RNF114 encodes a Di19 domain at its C-terminus, which has been implicated in recognition of MARylated histones (Longarini et al, 2023). Downstream of the Di19 domain is an annotated ubiquitin-interacting motif (UIM), which has been shown previously to bind polyUb chains (Fisher et al, 2003; Giannini et al, 2008). Considering the proximity of these two domains, we hypothesized that they might cooperate to bind MARUbe modifications. We used AlphaFold3 to model the tandem Di19-UIM module at the C-terminus of RNF114 together with two $Zn^{2+}$ ions, Ub, and NAD$^+$ as a proxy for ADPr (Fig. 3A). The resulting model scored with high confidence across the entire complex, except the Ub C-terminus that showed indications of flexibility (Fig. EV3A–C). The top 5 models provided by AlphaFold3 were all highly similar. Interestingly, the models placed Ub in such an orientation that its C-terminus extended towards the adenine-proximal ribose of NAD$^+$, where the 3' hydroxyl group is the site of NAD$^+$ and ADPr ubiquitylation by Deltex ligases (Zhu et al, 2022). Upon closer inspection, there appeared to be a distinct pocket within the Di19 domain that bound the NAD$^+$ moiety, with the nicotinamide group (which is replaced by a protein sidechain following ADP-ribosylation) exposed to solution and the only source of variation among the top 5 models. (Fig. EV3B). We identified two key coordination sites between the NAD$^+$ and RNF114. Namely, W181 and G182 that are contained in a loop forming the back end of the NAD$^+$-binding pocket, and R198 coordinating the diphosphate (Figs. 3B and EV3D). Y186 and E193 make additional hydrogen bonding contacts.

The C-terminal helix of RNF114 is a canonical UIM that coordinates Ub through the I44 patch in a mode shared among related ubiquitin-binding domains (UBDs) (Fig. 3C). V220, centered in the UIM of RNF114 is a key conserved residue among UIMs. The UIM helix is also flanked on either end with highly conserved acidic residues and S224. The UIM is extended at its N-terminus to also coordinate the Ub I36 patch with F201/F206 of RNF114, located in the short linker region between its Di19 and UIM domains. When examining the evolutionary conservation of the RNF114 Di19-UIM module using ConSurf, we found strong conservation in the NAD$^+$-binding site and on the Ub-interacting surface of the UIM helix (Fig. 3D). Our AlphaFold3 model

therefore suggests that RNF114 is poised to recognize MARUbylation marks through coordination of its Di19 domain and C-terminal UIM. Thus, as these modeling studies suggest that the two domains work in concert, we propose to rename the tandem Di19-UIM module the M̲ARU̲be-B̲inding D̲omain (M-UBD) and will refer to it as such going forward.

## M-UBD mutants impair Ub-ADPr binding

To validate the AlphaFold3 model of the RNF114:NAD$^+$:Ub complex, we rationalized point mutations of residues involved in binding ADPr, Ub, or Ub-ADPr. We observed no variation in the chromatograms of each mutant upon purification using size exclusion chromatography (Fig. EV3E), and thermal stability analysis showed no variation in the melting temperatures of the mutants compared to WT RNF114, suggesting the point mutations did not drastically alter the folding or stability of RNF114 (Fig. EV3F). We tested these different RNF114 mutants for binding to our fluorescent substrates ADPr$^T$ and Ub-ADPr$^T$ (Figs. 3E and EV3G). As a control for a canonical Ub modification, we also compared to the Ub-Lys$^T$ substrate generated previously (Geurink et al, 2012), in which Ub is isopeptide-linked to a TAMRA-labeled Lys-Gly dipeptide. Mutations within the ADPr-binding pocket, including W181D/G182E (hereinafter referred to as WG/DE) or R198E, nearly eliminated the ability of RNF114 to bind ADPr$^T$ (Figs. 3E, left and EV3G). A truncated version of RNF114, ΔUIM$^{ext}$, which ends immediately after the Di19 domain at residue 200, retained ADPr$^T$ binding comparable to WT RNF114. Though we could not reach saturation with our fluorescence polarization binding assay, we estimate the $K_d$ of RNF114 for ADPr$^T$ to be ~40 μM.

Using Ub-Lys$^T$ as a model for canonical Ub modifications, we could detect binding to RNF114 by fluorescence polarization but were again unable to calculate a $K_d$ for the interaction due to the high concentrations of enzymes needed, though we estimate it to be ~20 μM. We found that removal of the RING domain and associated disordered regions in RNF114 to create the M-UBD construct (residues 137–228) facilitated slightly better Ub binding than full-length RNF114 (Figs. 3E, middle and EV3G). The two ADPr-binding pocket mutants retained some of their ability to bind the Ub-Lys$^T$ substrate. Surprisingly, the F206E version of RNF114, which disrupts coordination of the Ub I36 patch, retained some Ub binding. The ΔUIM$^{ext}$ and V220E mutations significantly abrogated Ub binding, indicating the driving factor behind the noncovalent RNF114:Ub interaction lies at the center of the UIM.

Finally, we used Ub-ADPr$^T$ to study the binding of RNF114 mutants to MARUbe (Figs. 3E, right and EV3G). Consistent with a multimodal interaction, we detected much stronger binding to Ub-ADPr$^T$ compared to ADPr$^T$ and Ub-Lys$^T$. For WT RNF114, binding was saturated by 40 μM, and we calculated a $K_d$ of 12.8 μM (95% confidence interval: [7.74, 23.10]). As observed with the ADPr$^T$ or Ub-Lys$^T$ substrates, the WG/DE, V220E, and ΔUIM$^{ext}$ mutants of RNF114 were defective in their binding of Ub-ADPr$^T$. Similar to the Ub-Lys$^T$ substrate, the R198E mutant exhibited binding to Ub-ADPr$^T$ at nearly the same level as WT RNF114, indicating that R198E can overcome the inability to bind ADPr, whereas the WG/DE mutant could not. Another possibility is that R198 is not important for ADPr binding. Interestingly, we saw that RNF114 F206E retained binding of Ub-ADPr$^T$ at similar levels as WT RNF114, while its binding to Ub-Lys$^T$ was somewhat impaired. Together, these findings highlight the necessity of both the ADPr-

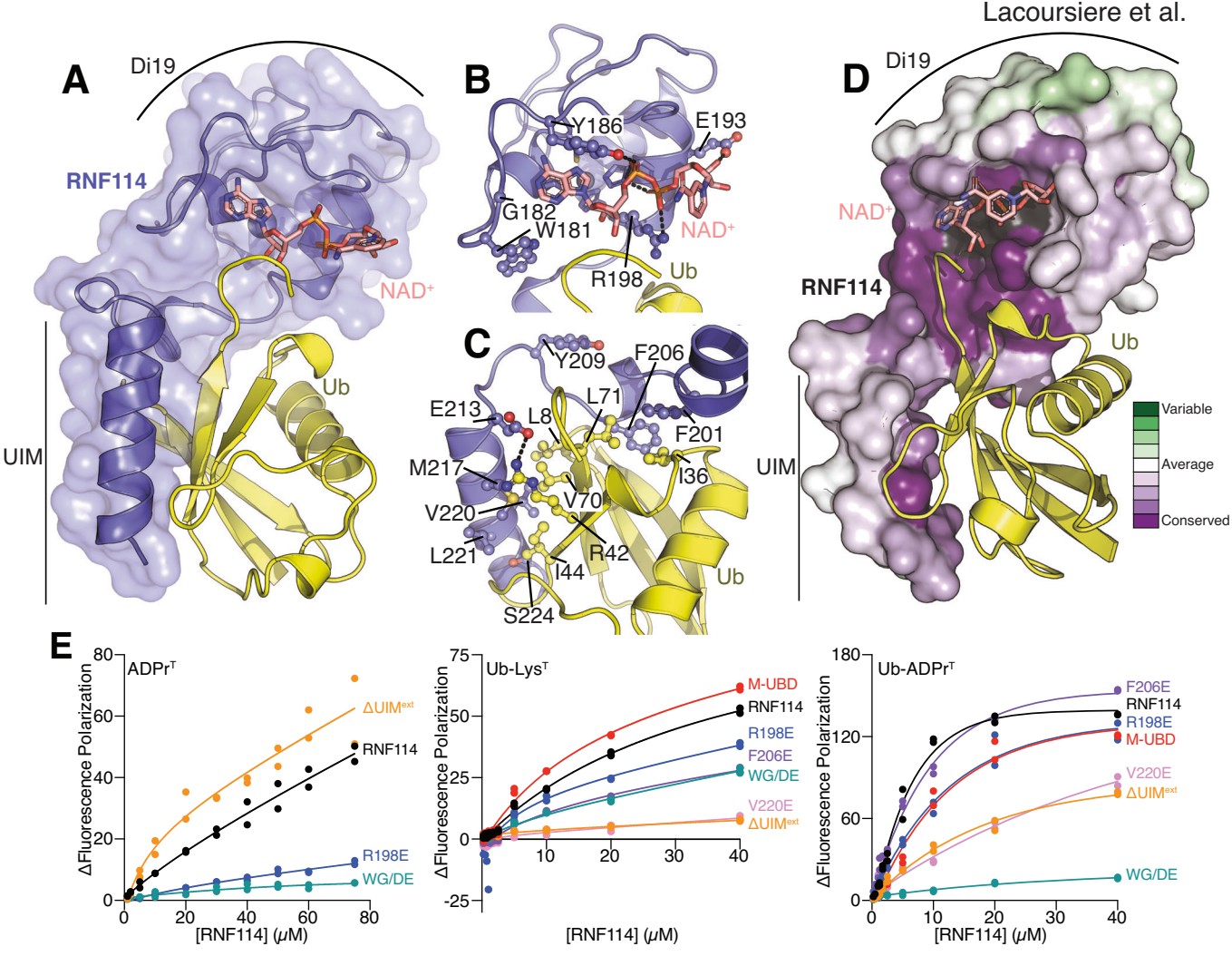

**Figure 3. RNF114 binds ADPr and Ub.**

(A) AlphaFold3 model of RNF114 Di19 and UIM domains (blue) in complex with two $Zn^{2+}$ ions (gray spheres), as well as $NAD^+$ (salmon) and Ub (yellow) to mimic a MARUbe modification. (B) Detailed view of the modeled interaction between the RNF114 Di19 domain and $NAD^+$. (C) Detailed view of the modeled interaction between the RNF114 UIM and Ub. (D) Consurf analysis showing the conservation within the Di19 domain and UIM of RNF114 throughout evolution. As shown, dark purple represents the conserved areas, while green areas are more variable. The surface shown in dark gray represents the highly conserved $Zn^{2+}$-binding structural residues in the Di19 domain. (E) Model-guided mutations in RNF114 alter binding to $ADPr^T$ (left), $Ub-Lys^T$ (middle), and $Ub-ADPr^T$ (right) substrates. Source data are available online for this figure.

and Ub-binding sites for a high-affinity interaction with MARUbe, but also that either binding site alone can maintain a weak affinity.

## RNF114 preferentially extends Ub-ADPr with polyUb

It is well accepted that many E3 ligases undergo rapid auto-ubiquitylation or catalyze the formation of free polyUb chains in vitro. We indeed observed free polyUb formation over the course of 1 h with RNF114 and Ub, confirming our construct was active (Fig. EV4A). We then sought to use our fluorescent substrates in a UbiReal experiment (Franklin and Pruneda, 2019) to examine activity towards $Ub-ADPr^T$ compared to $Ub-Lys^T$, $ADPr^T$, or $Ub-Lys^T$ in the presence of ADPr (Fig. 4A). In these experiments, unlabeled Ub was added to facilitate ubiquitylation of the

fluorescently labeled substrate. Under these conditions, the fluorescent Ub substrates can only behave as the "acceptor" ($Ub_A$) during polyUb formation due to modifications at their C-termini, while unlabeled Ub serves as the "donor" ($Ub_D$). Formation of polyUb exclusively with unlabeled Ub is possible, but blind to the fluorescence polarization readout of the UbiReal experiment. In the presence of ligase reaction components, we observed no increase in fluorescence polarization of $ADPr^T$ following the addition of ATP, indicating that RNF114 cannot form the initial Ub-ADPr. Conversely, we observed a robust increase in fluorescence polarization with substrates containing Ub, indicating extension of polyUb products. Remarkably, RNF114 demonstrated a strong preference for extending the $Ub-ADPr^T$ substrate compared to the $Ub-Lys^T$ substrate, regardless of whether unlabeled ADPr was

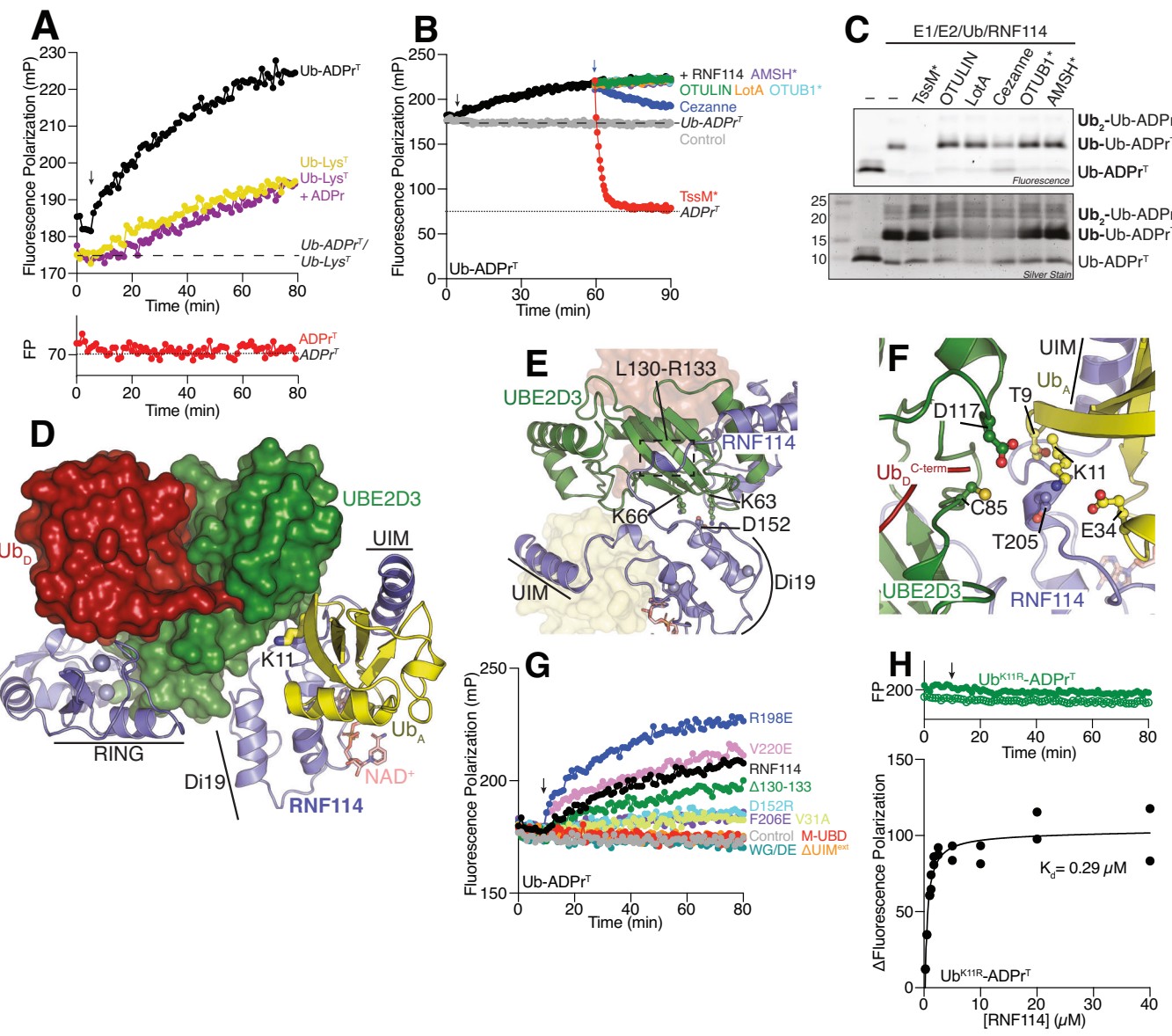

**Figure 4.  RNF114 extends K11 polyUb onto Ub-ADPr.**

(**A**) Representative UbiReal experiment showing the activity of RNF114 against the indicated substrates. The expected fluorescence polarizations of the substrates alone are indicated using dashed lines. Composition of the ligase experiment can be found in the Methods. These datasets are representative of $n = 3$ technical replicates for the Ub-ADPr$^T$ substrate or $n = 2$ technical replicates for the other substrates. (**B**) The products of a ligase reaction were subjected to a UbiCRest experiment to identify the linkage type present on the Ub-ADPr$^T$ substrate. Reactions were treated with the indicated deubiquitylase (added at the blue arrow), and their fluorescence polarization was monitored over time. Thirty minutes after DUB addition, the plate was removed from the instrument and samples were quenched in SDS sample buffer and (**C**) fluorescent products were visualized by SDS-PAGE. All DUBs except Cezanne were used at a concentration of 0.5 µM. Cezanne was used at 20 nM. (**D**) AlphaFold3 model of the complex between RNF114 (blue), NAD$^+$ (salmon), UBE2D3 (green), two copies of Ub (red/yellow), and five Zn$^{2+}$ ions (gray spheres). Each protein and the important regions in RNF114 are labeled. (**E**) RNF114 coordinates the E2 protein through backside binding (loop L130-R133) and charge interactions by D152. (**F**) RNF114 positions Ub$_A$ so the K11 sidechain extends into the active site of UBE2D3-Ub. To facilitate catalysis, a TEK box comprised of the Ub$_A$ T9 and E34 are within proximity to deprotonate K11 of the Ub$_A$. RNF114 T205 is also in close proximity. (**G**) A fluorescence polarization UbiReal experiment was used to evaluate the indicated regions of RNF114 and their potential roles in catalyzing the formation of Ub chains. (**H**) A representative UbiReal experiment of a full ligase reaction (solid circles) for RNF114 with a K11R version of the Ub-ADPr$^T$ (Ub$^{K11R}$-ADPr$^T$) substrate compared to the Ub$^{K11R}$-ADPr$^T$ substrate-only control (open circles). To examine binding, various concentrations of WT RNF114 were incubated with Ub$^{K11R}$-ADPr$^T$, and the data were collected as indicated in "Methods". The $K_d$ was derived using GraphPad Prism 10 using a nonlinear regression. The 95% confidence interval for the $K_d$ shown here is [0.20413, 0.7803]. Throughout all panels, the black arrow shows the point when ATP was added to the reaction. Source data are available online for this figure.

added to supplement the Ub-Lys$^T$ reaction. This finding suggests that the linkage between Ub and ADPr moieties contributes to the stronger binding and ligase activity observed for the Ub-ADPr substrate compared to unlinked Ub and ADPr: Simply occupying both binding sites with separate Ub and ADPr does not enhance ligase activity.

## RNF114 specifically extends K11 polyUb on a Ub-ADPr substrate

Confirming that RNF114 exhibited preferential ligase activity toward Ub-ADPr$^T$, we asked what type of Ub linkage RNF114 adds. To address this question, we used a UbiCRest experiment whereby the products of a ligase reaction (like those shown in Fig. 4A) were treated with a panel of deubiquitylases specific to distinct chain topologies (Hospenthal et al, 2015) (Fig. 4B). We selected deubiquitylases that are specific for Ub esters (TssM*; (Szczesna et al, 2024)), M1 polyUb (OTULIN; (Keusekotten et al, 2013)), K6 polyUb (LotA; (Warren et al, 2023)), K11 polyUb (Cezanne; (Mevissen et al, 2016)), K48 polyUb (OTUB1*; (Michel et al, 2015)), or K63 polyUb (AMSH*; (Michel et al, 2015)). As we had previously observed K11 extension of MARUbe on PARP7 and PARP10 from cells (Bejan et al, 2025), and observed reduced polyUb extension on PARP7 MARUbe following RNF114 knockdown (Fig. 1C,D), we hypothesized that RNF114 was extending K11 polyUb onto Ub-ADPr. Initial testing of Cezanne against the RNF114 ligase products demonstrated that high concentrations of this enzyme were able to cleave the Ub-ADPr ester linkage, as evident by the fluorescence polarization returning to values <100, consistent with the observed fluorescence polarization of ADPr$^T$ (Fig. EV4B). We therefore optimized a concentration of 20 nM Cezanne, which did not cleave the Ub-ADPr ester linkage after 30 min. Importantly, all of the other polyUb deubiquitylases included in our UbiCRest panel are exclusive to their polyUb substrates and cannot cleave other types of Ub modification (Hospenthal et al, 2015; Warren et al, 2023). The UbiCRest experiment demonstrated that the only deubiquitylase other than Cezanne that cleaved RNF114 ubiquitylation products was the Ub esterase TssM*. Notably, Cezanne treatment returned the fluorescence polarization to ~180, in good agreement with values for Ub-ADPr$^T$ alone (Fig. 4B). Treatment with TssM* returned the fluorescence polarization to ~70, in good agreement with the observed polarization of ADPr$^T$ alone, indicating that TssM* had likely cleaved the entire polyUb chain from ADPr. Indeed, when we examined products of the UbiCRest experiment by SDS-PAGE and in-gel fluorescence, we observed a strong signal for Ub-Ub-ADPr$^T$ produced by RNF114 (Fig. 4C). In the presence of TssM*, the fluorescence signals corresponding to Ub-Ub-ADPr$^T$ and Ub-ADPr$^T$ were eliminated, consistent with the release of ADPr$^T$. Upon silver staining, we observed an intense protein band correlating to diUb, confirming our suspicions that TssM* had cleaved the entire polyUb from ADPr$^T$. Cezanne treatment decreased the intensity of the Ub-Ub-ADPr$^T$ band by ~50% evidenced by the decrease in fluorescence polarization to ~190 mP (Fig. 4B) and band intensity from evaluating this reaction on SDS-PAGE, and we observed the reappearance of a fluorescent band at the molecular weight of Ub-ADPr$^T$. We expect that longer timepoints would allow for complete cleavage of the ubiquitylation products by Cezanne and fluorescence polarization return to ~180 mP (that of Ub-ADPr$^T$). The other tested deubiquitylases showed no decrease in fluorescence of the Ub-Ub-ADPr$^T$ species, despite their concentrations being 25x higher than that of Cezanne. We then completed the same

UbiCRest experiment for ligase products using Ub-Lys$^T$ to address whether specificity was retained (Fig. EV4C). For products generated with Ub-Lys$^T$, we again observed robust cleavage with Cezanne, indicating the specific formation of K11 polyUb. As expected in this experiment, TssM* did not cleave the products, as there was no ester-linked Ub present in the reaction. Taken together, these findings support the hypothesis that RNF114 specifically extends K11 polyUb on its preferred target of Ub-ADPr.

## Molecular basis for RNF114 extension of K11 polyUb on Ub-ADPr

To examine the molecular basis for K11 extension of MARUbe by RNF114, we used AlphaFold3 to model the full transferase complex containing full-length RNF114, UBE2D3, NAD$^+$, two copies of Ub, and five Zn$^{2+}$ ions (Fig. 4D). Remarkably, the model had high confidence throughout most of the complex (Fig. EV4D,E). Outside of a disordered region N-terminal to the RNF114 RING domain, only one region of RNF114, between residues 128–134 that cross over the backside of the E2 protein, was scored as low confidence. The top 5 scoring models from AlphaFold3 aligned nearly perfectly, with slight variation at the N-terminus of RNF114 and in the positioning of the NAD$^+$ nicotinamide group (Fig. EV4F). We observed a canonical E2:RING interface whereby helix α1, and loops 4 and 7 of UBE2D3 mediate the interaction (Fig. EV4G). The RING:E2 interface is extended by an additional RNF114 zinc finger that binds UBE2D3 helix 1 analogously to a previously determined crystal structure of RNF125 (Middleton et al, 2023). We also observed UBE2D3 backside coordination by RNF114 residues 128–134 and a potential charge interaction by D152 (Fig. 4E). One copy of Ub was found to mimic a thioester intermediate with UBE2D3, wherein the Ub C-terminus extended toward the UBE2D3 C85 active site, and the two proteins formed a "closed" conformation of E2~Ub observed in RING-dependent priming of Ub transfer (Dou et al, 2012; Plechanovova et al, 2012; Pruneda et al, 2012). The second copy of Ub, together with NAD$^+$, bound into the M-UBD module at the RNF114 C-terminus in a manner identical to our previous modeling. Consistent with our biochemical analysis of RNF114 specificity, further examination of this complex revealed the K11 sidechain of the MARUbe was around 6 Å away from and pointed directly towards the active site of UBE2D3, supported by residues from both the E2 and RNF114 (Fig. 4F). Thus, the full transferase complex model suggests K11-specific ubiquitylation of MARUbe. We designed a series of point mutations to validate the model, which were tested in our UbiReal experiments (Figs. 4G and EV4H). As expected, none of the RNF114 mutants tested could ubiquitylate the ADPr$^T$ substrate, and many mutants showed no activity towards Ub-Lys$^T$ either in the absence or presence of unlabeled ADPr (Fig. EV4H). We observed ubiquitylation of the Ub-Lys$^T$ substrate with the R198E or Δ130–133 versions of RNF114, suggesting that (1) Ub binding to the UIM can overcome charge differences in the ADPr-binding pocket; and (2) the backside binding of UBE2D3 mediated by residues 130–133 of RNF114 does not strongly influence ubiquitylation. Other E2-coordinating mutants, V31A or D152R were completely defective in their ubiquitylation of Ub-Lys$^T$. All E2-binding mutants of RNF114 (V31A, Δ130–133, and D152R) retained some ubiquitylation of the Ub-ADPr$^T$ substrate, where the Δ130–133 mutant exhibited ligase activity comparable to WT

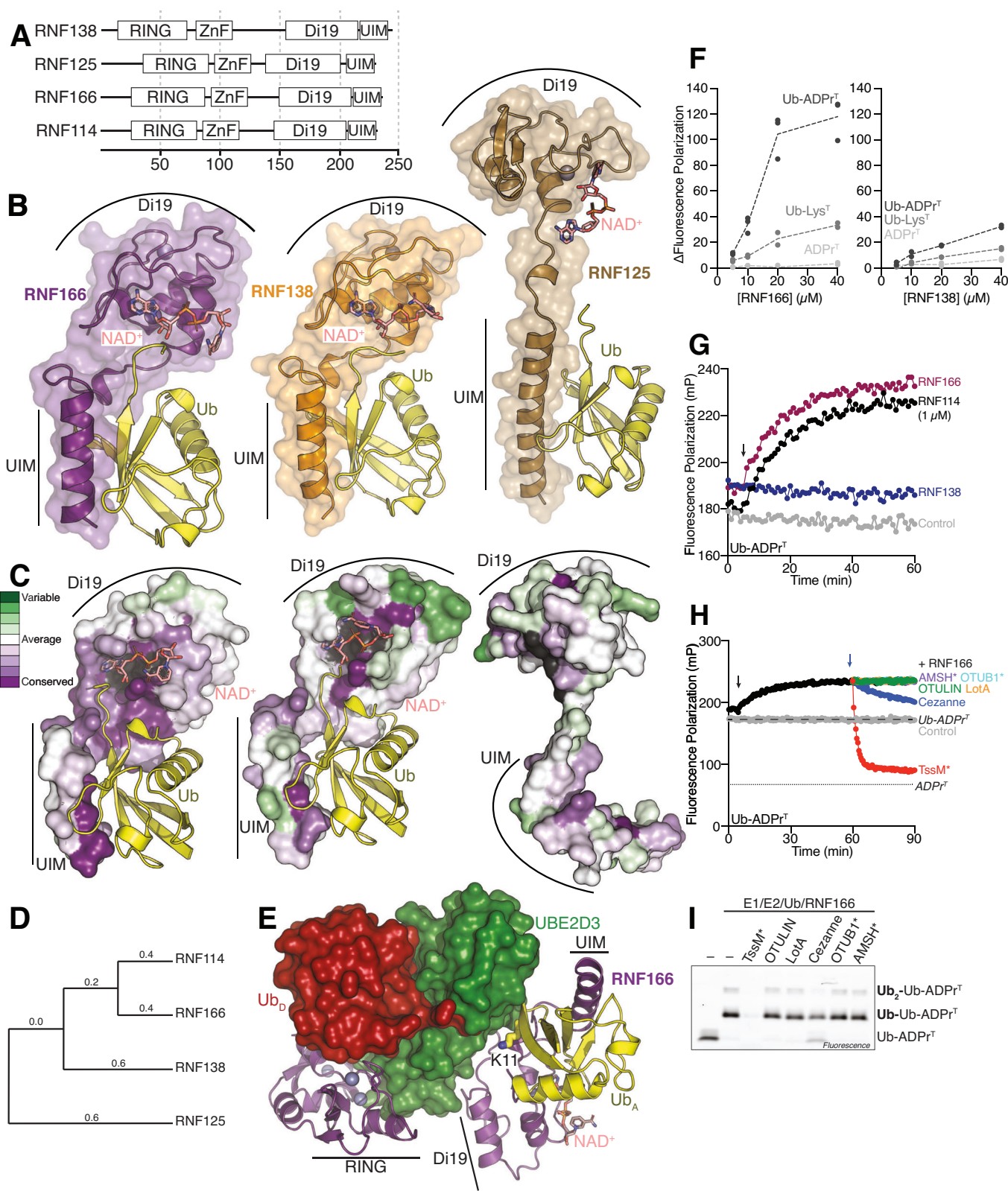

**Figure 5. RNF114 belongs to a family of MARUbe-Targeted Ligases (M-UTLs).**

(A) Schematic representation of the domain organization for RNF114, RNF125, RNF138, and RNF166, other Di19 domain- and UIM-containing E3 ligases. (B) AlphaFold3 models depicting the M-UBD of the indicated E3 ligases in complex with two $Zn^{2+}$ ions, as well as $NAD^+$ and Ub. (C) Consurf analysis of the indicated E3 ligases ordered as above showing evolutionary conservation (purple) or variability (green). (D) Evolutionary divergence tree for the human versions of RNF114, RNF125, RNF138, and RNF166. (E) AlphaFold3 model of RNF166 (violet), UBE2D3 (green), $NAD^+$ (salmon), two copies of Ub (red/yellow), and five $Zn^{2+}$ ions (gray spheres). Shown is the positioning of the $Ub_A$ K11 sidechain extending towards the UBE2D3 active site in an orientation similar to the model for RNF114. (F) RNF166 and RNF138 preferentially bind $Ub-ADPr^T$ over $Ub-Lys^T$ and $ADPr^T$ substrates. (G) Representative UbiReal experiment for the indicated E3 ligases with the $Ub-ADPr^T$ substrate. The control represents the baseline fluorescence polarization of the substrate without any other proteins. In this experiment, RNF114 is 1 μM, and the other ligases are 5 μM. (H) The products of an RNF166 ligase reaction (black dataset) were subjected to a UbiCRest experiment with the indicated deubiquitylases added at 60 min, indicated by the blue arrow. All DUBs except Cezanne were used at 0.5 μM, and Cezanne was used at 20 nM. (I) Products of the UbiCRest experiment were evaluated using in-gel fluorescence following SDS-PAGE. Throughout the panels, the black arrow marks the point when ATP was added to the reactions. Source data are available online for this figure.

RNF114, and the V31A and D152R mutants reduced activity (Fig. 4G). The F206E mutant of RNF114 displayed ubiquitylation activity similar to V31A and D152R. Surprisingly, we observed robust ubiquitylation with the V220E and R198E mutants of RNF114, suggesting that despite compromised binding of either the $ADPr^T$ or $Ub-Lys^T$ substrates, RNF114 ligase activity does not require tight binding of the MARUbe to facilitate its ubiquitylation. There were three mutants that did not possess any ligase activity toward the $Ub-ADPr^T$ substrate: WG/DE, M-UBD, and $\Delta UIM^{ext}$. The lack of activity was expected for the WG/DE and M-UBD mutants due to their inability to bind $Ub-ADPr^T$ and the lack of the RING domain, respectively. However, we were intrigued to see that the $\Delta UIM^{ext}$ version of RNF114 failed to ubiquitylate the $Ub-ADPr^T$ substrate despite its only moderately impaired binding of the substrate (Fig. 3E, right). It is possible that, while these mutants can still bind Ub-ADPr, an interaction within the UIM is required to orient Ub into the E2 active site. Consistent with this, the UIM of RNF166 was shown to be required for efficient ubiquitylation of TNKS (Perrard and Smith, 2023). Taken together, these data not only validate the structural model but also show that RNF114 ligase activity can overcome defective binding due to mutation in either the Ub or ADPr-binding sites.

Because the structural modeling, coupled with our earlier UbiCRest data, suggested tight specificity for K11 ubiquitylation, we sought to test whether RNF114 could produce any other polyUb product. To address this, we used a K11R mutant of Ub and the methodology described in Fig. 2 to form $Ub^{K11R}-ADPr^T$ (Fig. EV4I). We validated the binding of this substrate to WT RNF114 and were surprised to see much tighter binding compared to the $Ub-ADPr^T$ substrate ($K_d = 290$ nM) (Fig. 4H). In a UbiReal experiment, RNF114 failed to extend any Ub onto the $Ub^{K11R}-ADPr^T$ substrate, even after 1 h (Fig. 4H). This result again confirms that RNF114 specifically extends K11 linkages on the Ub-ADPr substrate. Furthermore, none of the structure-guided RNF114 mutants could extend Ub onto the $Ub^{K11R}-ADPr^T$ substrate (Fig. EV4J), confirming that none of the retained activity observed with some mutants was due to formation of different polyUb linkages. Taken together, these data further indicate that RNF114 synthesizes exclusively K11 linkages on a Ub-ADPr substrate.

## RNF114 belongs to a family of MARUbe-targeted ligases (M-UTLs)

The initial identification of RNF114 as a Di19 domain-containing protein also identified Di19 domains in the E3 ligases RNF125, RNF138, and RNF166 (Fig. 5A) (Giannini et al, 2008). A sequence alignment of these Di19-containing E3 ligases showed a conservation

of the $Zn^{2+}$- and E2-coordinating residues, as well as M-UBD architecture (Fig. EV5A). One key difference we noticed was the length of the linker connecting the RING-ZnF ligase module to the M-UBD, where RNF114 and RNF166 had the same linker length, while the linkers in RNF125 and RNF138 were shorter or longer, respectively. Interestingly, we observed a conserved ADPr-binding site in RNF114, RNF138, and RNF166 consisting of a WGD motif, which was missing in RNF125. Specifically, the residues that create part of the backside of the ADPr-binding pocket (W181/G182) were not conserved in RNF125. Instead, RNF125 possesses D178/E179, which based on our mutagenesis work with RNF114, diminishes ADPr binding. Key residues in the UIM region remain conserved, including the charged residues on the N-terminal side of the helix and the polar serine near the C-terminus.

Based on this sequence alignment, we wondered if these other ligases would show activity for a MARUbe substrate. We used AlphaFold3 to model RNF125, RNF138, or RNF166 with $NAD^+$ and Ub to evaluate possible binding to the M-UBD (Fig. 5B). For RNF138 and RNF166, which had higher similarity with RNF114, AlphaFold3 modeled the complexes with high confidence (Fig. EV5B). Consistent with our model of RNF114 (Figs. 3 and EV3), the nicotinamide group was observed in several orientations extending into solution (Fig. EV5C). Consistent with the lack of conservation in the ADPr-binding pocket, AlphaFold was unable to produce a high-confidence model for RNF125 in complex with $NAD^+$ and Ub. When examined with ConSurf, RNF166, RNF138, and RNF125 each had a different level of evolutionary conservation (Fig. 5C). Notably, RNF166 possessed a nearly continuous conserved patch stretching from the ADPr-binding site to the Ub-binding face of the UIM. RNF138 demonstrated regions of conservation in the ADPr-binding site and toward the C-terminus of the UIM with regions of variation dispersed throughout the M-UBD. RNF125, however, demonstrated little conservation including throughout the ADPr-binding site, supporting its evolutionary divergence from the other ligases studied here (Fig. 5D). Together, these findings suggest that RNF125 might not bind $NAD^+$. We then attempted to model full transferase complexes for RNF125, RNF138, and RNF166 in AlphaFold3. For RNF125 and RNF138, AlphaFold3 failed to produce high-confidence models, likely a result of their distinct linker lengths. Comparatively, in the transferase complex modeled for RNF166, we again observed the MARUbe K11 sidechain extending towards the active site of UBE2D3~Ub (Fig. 5E). We also noticed that the model exhibited high confidence throughout the majority of the complex, and the top 5 scoring models were very similar (Fig. EV5D,E).

We aimed to validate the AlphaFold3 models with Ub-ADPr-binding experiments for each of the three ligases; however, we

could not produce soluble RNF125 for these experiments. Similar to RNF114, and as expected from the AlphaFold3 models, both RNF166 and RNF138 showed selective binding for Ub-ADPr$^T$ over ADPr$^T$ and Ub-Lys$^T$, as demonstrated by much higher changes in fluorescence polarization (Fig. 5F). The binding experiments with RNF138 showed considerably smaller changes in fluorescence polarization, which could be explained by a lower affinity and/or higher degree of conformational heterogeneity. These binding assays corroborate the AlphaFold3 models and sequence alignment, showing that RNF114, RNF166, and RNF138 all have functional M-UBD modules capable of reading MARUbe modifications.

We next evaluated the ligase activity of RNF138 and RNF166 in a UbiReal experiment with our fluorescent substrates. Surprisingly, only RNF166 was able to extend polyUb onto Ub-ADPr$^T$ but required five times higher concentration than RNF114 (Fig. 5G). RNF166 was also able to ubiquitylate Ub-Lys$^T$ at these higher concentrations, though we observed a strong preference for the Ub-ADPr$^T$ substrate (Fig. EV5F). We performed a UbiCRest analysis to determine which chain topology RNF166 was building onto Ub-ADPr$^T$ (Fig. 5H). Upon treating the ligase reaction with our panel of specific deubiquitylases, we observed that only TssM* and Cezanne were able to cleave the ubiquitylated Ub-ADPr$^T$, similar to RNF114. TssM* cleaved the products down close to the expected polarization of ADPr$^T$ (~85), as shown by the signal decrease below the Ub-ADPr$^T$ control. Cezanne, the K11-specific deubiquitylase, only partially cleaved the extended Ub-ADPr$^T$, returning the polarization to ~200. These samples were then evaluated on SDS-PAGE with in-gel fluorescence and we observed that the main product was Ub-Ub-ADPr$^T$ and a minor amount of Ub$_2$-Ub-ADPr$^T$ (Fig. 5I). We also conducted a UbiCRest experiment on a RNF166 ligase reaction utilizing Ub-Lys$^T$ as a substrate and observed the same trends as RNF114 (Fig. EV5G,H). Here, RNF166 synthesized longer linkages on the Ub-Lys$^T$ substrate that were only cleaved by the K11-specific Cezanne. Taken together, we show that M-UBD-containing E3 ligases can be considered a unique family of MARUbe-Targeted Ub Ligases, or M-UTLs, that can read MARUbe modifications and write new extensions of polyUb which, at least for RNF114 and RNF166, are specifically linked via K11.

# Discussion

Here, we show that a family of E3 Ub ligases containing a tandem Di19-UIM module can specifically recognize ubiquitylated mono-ADPr (MARUbe). We extensively evaluate the structure and function of one of these family members, RNF114, and decipher the mechanism by which RNF114 builds a K11-linked polyUb onto MARUbe. We show that RNF114 has a strong preference for the MARUbe substrate compared to ADPr or Ub alone (Figs. 3E and 4A). We highlight how RNF114 positions MARUbe so that K11 reaches towards the active site of an incoming E2~Ub conjugate (Fig. 4D,F), and we extend our observations of RNF114 to a family of Ub ligases containing a tandem Di19-UIM. We found that at least 2 other members (RNF138, RNF166) preferentially bind MARUbe compared to ADPr or Ub alone, and one, RNF166, catalyzes the K11-linked ubiquitylation of MARUbe. We therefore suggest renaming the tandem Di19-UIM the MarUbe-binding domain (M-UBD) and the family of ligases containing RNF114 and RNF166 MARUbe-targeted ligases (M-UTLs). Due to their inability to extend polyUb onto MARUbe, RNF125 and RNF138

remain as candidate M-UTL proteins. RNF125 and RNF138 appear to have arisen more recently in evolution compared to RNF114 and RNF166, as both appear only in tetrapods while RNF114 is present in jawed vertebrates and RNF166 is more widely found across vertebrates. Interestingly, the appearance of M-UTLs in vertebrates overlaps with an expansion of PARP enzymes (Citarelli et al, 2010).

Although all members of the M-UTL family except RNF125 possess an M-UBD, we only observed ubiquitylation of Ub-ADPr with RNF114 and RNF166, suggesting differences in catalytic activity between each of the family members. One obvious source of variation is the linker length between the MARUbe reader and writer modules. RNF114 and RNF166 have the same linker length, while RNF138 is longer and RNF125 is shorter. The linker length could directly relate to the conformation of the writer and reader modules after MARUbe binding, whereby only the RNF114 and RNF166 complexes have the correct structural organization to facilitate MARUbe ubiquitylation. Despite similar activity in vitro, RNF166 did not extend K11-linked ubiquitylation onto PARP7 MARUbe in cells (Fig. 1C). This finding suggests that RNF114 and RNF166 are not functionally redundant and therefore might have evolved to target unique MARUbylated substrates. Interestingly, RNF125 is the only related ligase that does not retain a conserved M-UBD (Fig. 5C), which could suggest that either the RNF125 M-UBD is degenerate or that it is rapidly evolving toward a substrate other than MARUbe.

We and others have shown that various PARP proteins (PARP1, PARP7, PARP10, TNKS) are MARUbylated in cells, and that the MARUbe modification is extended with K11 polyUb (Bejan et al, 2025; Kolvenbach et al, 2025; Perrard et al, 2025). A current model suggests that MARUbylation could stabilize a substrate by blocking PAR extension and preventing downstream PARdU, but it remains unclear if this outcome is consistent across all MARUbylated substrates (Perrard et al, 2025). PARPs are known to MARylate the sidechain of various residues, including Glu/Asp, Ser/Thr, and Cys, and the lability of the bond changes with each. For PARP10, we have shown that MARUbylation occurs specifically on Glu/Asp residues (Bejan et al, 2025). Meanwhile, TNKS and PARP1 were suggested to be MAR-Ubylated on serine (Kolvenbach et al, 2025; Perrard et al, 2025). Together, these findings suggest the lability and transient nature of MARUbylation, indicating the species might serve as an intermediate in some undescribed signaling pathway. In our siRNA knockdown experiments, we observed decreased K11-linked MARUbylation concomitant with an increase in PARP7 levels following DTX2 and RNF114 knockdown (Fig. 1D). This suggests a role for K11-linked MARUbylation in either stabilizing PARP7 protein or affecting its compartmentalization, which might not be detectable under our current lysis conditions. While this manuscript was in review, an independent report confirmed RNF114 activity towards Ub-ADPr in vitro (Kloet et al, 2025). While some reports suggest that MARUbylation contributes to protein stabilization, it is imperative to continue studies on the outcomes of this novel dual PTM to fully delineate the outcome of MARUbylation vs. canonical substrate ubiquitylation (Perrard and Smith, 2023). PARP7, for example, is modified with both MARUbylation as well as canonical, Lys-linked ubiquitylation (Fig. 1C; α-HA blots; beads vs. supernatant), and future studies are needed to carefully dissect the functional outcomes of these different forms of Ub modification.

The abundant crosstalk between PARPs and ubiquitylation at sites of DNA damage provides an obvious axis for therapeutic intervention. PARP inhibitors such as olaparib, niraparib, ruca-parib, and talazoparib are currently being used to treat various

cancers (Del Campo et al, 2019; Matulonis et al, 2016; Moore et al, 2018; Swisher et al, 2017), and the natural product nimbolide has been shown to target RNF114, preventing its ubiquitylation and degradation of PARP1 (Li et al, 2023). The multilayered regulation of a substrate through MARUbylation provides an additional opportunity for therapeutic development and the possibility of co-treatment with current PARP inhibitors. As M-UTL activity occurs downstream of PARP activity, targeting the M-UTL family of E3 ligases through their ability to bind and read marks of MARUbylation will provide an added layer of specificity unreachable solely with PARP inhibitors.

Overall, the work presented here structurally and functionally demonstrates specific K11 polyUb extension of MARUbe modifications by a family of related E3 ligases. Additional layers of regulation for this complex PTM, including at the level of M-UTL activity, will be an interesting area of future work. Within RNF114, for example, there is strong evidence for phosphorylation at Tyr116, Tyr186, and Tyr209, which lie at interfaces with the E2, ADPr, and Ub, respectively (Figs. 3B,C and EV4G) (Hornbeck et al, 2015). Furthermore, the functional rationale for K11 polyUb specificity remains a fascinating mystery. While K11 linkages have previously been implicated in cell cycle regulation by the APC/C, they are in the context of heavily branched K11/K48 polyUb chains that accelerate proteasomal degradation (Wu et al, 2010; Yau et al, 2017). Alongside PARdU, MARUbylation represents a complicated component of the PARP/Ub axis, and future work will expand how these heavily regulated PTMs synergize in cellular signaling.

# Methods

### Reagents and tools table

| Reagent/resource | Reference or source | Identifier or catalog number |
|---|---|---|
| **Experimental models** | | |
| HEK 293T | ATCC | CRL-3216 |
| **Recombinant DNA** | | |
| pET28b-DTX2$^{389\text{-}622}$ (*H. sapiens*) | Bejan et al, 2025 | N/A |
| pET28a-UBE2D3 (*H. sapiens*) | Gift from R. Klevit | N/A |
| pET17b-Ub (*H. sapiens*) | Gift from D. Komander | N/A |
| pET21d-UBE1 (*H. sapiens*) | Berndsen & Wolberger, 2011 | Addgene #34965A |
| pETM30-TssM and TssM* (*B. pseudomallei*) | Szczesna et al, 2024 | N/A |
| pOPINS-USP21$^{196\text{-}565}$ (*H. sapiens*) | Ye et al, 2011 | Addgene #61585 |
| pOPINE-Cezanne$^{129\text{-}438}$ (*H. sapiens*) | Mevissen et al, 2016 | N/A |
| pOPINB-LotA-N$^{1\text{-}300}$ (*L. pneumophila*) | Warren et al, 2023 | N/A |
| pOPINB-OTUB1* (*H. sapiens*) | Michel et al, 2015 | Addgene #65441 |
| pOPINB-AMSH* (*H. sapiens*) | Michel et al, 2015 | Addgene #66712 |

| Reagent/resource | Reference or source | Identifier or catalog number |
|---|---|---|
| pOPINB-Otulin (*H. sapiens*) | Keusekotten et al, 2013 | Addgene #61464 |
| HA-Ub | Gift from A. Barnes | |
| pOPINB-RNF114 WT | This study | N/A |
| pOPINB-RNF166 | This study | N/A |
| pOPINS-3C-RNF138 | This study | N/A |
| pEGFP-C1-WT-PARP7 | Rodriguez et al, 2021 | N/A |
| GST-AF1521 | Gift from M. Nielsen | N/A |
| **Antibodies** | | |
| Mono-ADP-Ribose AbD33204 (1:250 = 2 µg/ml) | Bio-Rad | HCA354 |
| GFP (1:2000) | Proteintech | pabg1 |
| RNF114 (1:1000) | Genetex | GTX107046 |
| DTX2 (1:500) | Thermo Fisher Scientific | PA5-60164 |
| Tubulin (DM1A) (1:2000) | Cell signaling technology | 3873 |
| HA-tag (C29F4) (1:1000) | Cell signaling technology | 3724 |
| Peroxidase AffiniPure™ Goat Anti-Rabbit IgG (H + L) (1:10,000) | Jackson ImmunoResearch | 111-035-144 |
| Goat anti-Mouse IgG (H + L) Secondary Antibody, HRP (1:5000) | Invitrogen | 62-6520 |
| **Oligonucleotides and other sequence-based reagents** | | |
| siRNA NT | Horizon Discovery | D-001810-10-05 |
| siRNA RNF114 | Horizon Discovery | L-007024-00-0005 |
| siRNA DTX2 | Horizon Discovery | L-007114-00-0005 |
| siRNA RNF166 | Horizon Discovery | L-007119-00-0005 |
| PCR primers | This study (Table EV1) | |
| **Chemicals, enzymes, and other reagents** | | |
| DMEM, high glucose | Gibco | 11965118 |
| Fetal bovine serum | Sigma-Aldrich | F0926 |
| GlutaMAX™ Supplement | Gibco | 35050061 |
| Sodium pyruvate (100 mM) | Gibco | 11360070 |
| Basticidin S HCl | Corning | 30100RB |
| Doxycycline hyclate | Thermo Scientific chemicals | 446060050 |
| jetOPTIMIUS® DNA transfection reagent | Polypolus-transfection | 101000025 |
| DharmaFECT 1 Transfection Reagent | Horizon Discovery | T-2001-02 |
| Bovine serum albumin | Sigma-Aldrich | A9647 |
| TCEP | Thermo scientific | 20490 |
| cOmplete™, EDTA-free Protease inhibitor cocktail | Roche | 11873580001 |
| BioRad Protein assay dye reagent concentrate | Bio-Rad | 5000006 |

| Reagent/resource | Reference or source | Identifier or catalog number |
|---|---|---|
| Phthal 01 (pan-PARP inhibitor) | Rodriguez et al, 2021 | N/A |
| PR-619 | Medchemexpress | HY-13814 |
| 4–20% mini-PROTEAN® TGX™ precast protein gels | Bio-Rad | 4561096 |
| Nitrocellulose transfer kit | Bio-Rad | 1704270 |
| Carnation instant nonfat dry milk | Nestle | 12428935 |
| UltraPure™ sodium dodecyl sulfate (SDS) | Invitrogen | 15525017 |
| SuperSignal™ West Pico PLUS Chemiluminescent Substrate | Thermo Scientific | 34578 |
| SuperSignal™ West Femto Maximum Sensitivity Substrate | Thermo Scientific | 34095 |
| ChromoTek GFP-Trap® Magnetic Agarose | Proteintech | gtma |
| Urea | Fisher Scientific | BP169 |
| Hydroxylamine solution | Sigma-Aldrich | 438227 |
| ATP | Sigma-Aldrich | A2383 |
| HisPur™ Cobalt Resin | Thermo Scientific | 89965 |
| RBN2397 | Medchemexpress | HY-136174 |
| Compound 8 (OP-ADPr) | This study | N/A |
| NucleoSpin RNA Plus Kit | Macherey-Nagel | NC1749027 |
| ProtoScript II First Strand cDNA Synthesis Kit | NEB | E6560S |
| PowerUp SYBR green Master Mix | ThermoFisher Scientific | A25742 |
| $MgCl_2$ | Fisher Scientific | M33-500 |
| Dithiothreitol (DTT) | Fisher Scientific | BP172-25 |
| HEPES | Fisher Scientific | BP310-1 |
| NaCl | Fisher Scientific | S271-10 |
| β-mercaptoethanol | Sigma-Aldrich | M3148 |
| Tryptone | Fisher Scientific | BP9726-2 |
| Yeast Extract | Fisher Scientific | BP9727-2 |
| Isopropyl β-D-1-thiogalactopyranoside (IPTG) | GoldBio | 367-93-1 |
| SigmaFast protease inhibitor cocktail | Sigma-Aldrich | S8830-20TAB |
| PMSF | RPI | P20270-5.0 |
| Lysozyme | Thermo Scientific | J60701.06 |
| DNase | Sigma-Aldrich | 10104159001 |
| TAMRA-azide | Vectors Laboratories | CCT-AZ109-5 |
| Tris(3-Hydroxypropyltriazolylmethyl)amine (THPTA) | Sigma-Aldrich | 762342-100MG |
| $CuSO_4·5H_2O$ | Fisher Scientific | S73268 |
| Sodium ascorbate | Combi-Blocks | QE-4378 |
| Sodium acetate trihydrate | Fisher Scientific | S209-500 |

| Reagent/resource | Reference or source | Identifier or catalog number |
|---|---|---|
| SYPRO Orange Protein gel stain | Sigma-Aldrich | S5692-500ul |
| **Software** | | |
| GraphPad Prism 10.2.1 | https://www.graphpad.com/ | |
| Image Lab | https://www.bio-rad.com/en-us/product/image-lab-software?ID=KRE6P5E8Z | |
| AlphaFold3 Web interface | Abramson et al, 2024 | |
| PAE Viewer | Elfmann and Stulke, 2023 | |
| Consurf Web interface | Ashkenazy et al, 2016; Ashkenazy et al, 2010; Celniker et al, 2013; Glaser et al, 2003; Landau et al, 2005 | |
| Jalview | Waterhouse et al, 2009 | |
| MEGA12 interface | Kumar et al, 2024; Stecher et al, 2020 | |
| MARS data analysis | BMG Labtech | |
| ClarioStar Control | BMG Labtech | |
| Xcalibur | Thermo Scientific | |
| Freestyle | Thermo Scientific | |
| PyMol | The PyMOL Molecular Graphics System, Version 3.0 Schrödinger, LLC | |
| Unicorn | Cytiva | |
| **Other** | | |
| Trans-Blot® Turbo™ transfer system | Bio-Rad | 1704150 |
| ChemiDoc MP Imaging System | Bio-Rad | 12003154 |
| AKTA Pure | Cytiva | |
| Superdex75 Increase 10/300 GL | Cytiva | |
| HiLoad 16/600 Superdex75 | Cytiva | |
| CLARIOstar microplate reader | BMG Labtech | |
| Microplate, f-bottom black 384-well | Greiner Bio-One | 784900 |
| MicroAmp Fast 96-well reaction plates | Applied Biosystems | 4346907 |
| QuantStudio 3 Real-Time PCR system | Applied Biosystems | |
| LTQ Velos Pro Linear ion, C18 trap | Thermo Scientific | |
| Poroshell 300SB-C18 RP-HPLC column | Agilent | 661750-902 |

## Cell culture

Cells were cultured at 37 °C and 5% $CO_2$. HEK 293T (ATCC, CRL-3216) were cultured in DMEM (Gibco, 11965118) supplemented with 10% FBS (Sigma-Aldrich, F0926), 1× GlutaMAX (Gibco, 35050061), and 1 mM sodium pyruvate (Gibco, 11360070).

## siRNA knockdown and GFP-PARP7 overexpression

HEK 293T cells were reverse transfected with 50 nM total siRNA (Horizon Discovery) targeting RNF114 (cat no. L-007024-00-0005), DTX2 (cat no. L-007114-00-0005), RNF166 (cat no. L-007119-00-0005), or a non-targeting control (cat no. D-001810-10-05), using DharmaFECT 1 Transfection Reagent (Horizon Discovery, cat no. T-2001-02). After 48 h, cells were then co-transfected with GFP-PARP7 and HA-Ub plasmids using jetOPTIMUS® DNA transfection Reagent (Polyplus-transfection, cat no. 101000025) for 24 h (media was exchanged 4 h post-transfection). RBN2397 was used at 300 nM for 18 h following the media swap. Cells were then lysed and immunoprecipitated as described below.

## GFP immunoprecipitation for on-bead enzyme/chemical treatment assay (MARUbylation assay)

GFP-Trap® Magnetic beads (Proteintech, gtma) were washed twice with cell lysis buffer (CLB; 50 mM HEPES pH 7.4, 150 mM NaCl, 1 mM $MgCl_2$, 1% Triton X-100) and added to lysates at ~300–500 µg total protein/10 µl bead slurry. The beads were rotated for 2 h at 4 °C, then washed once with CLB, twice with 7 M Urea/1% SDS in PBS, once with 1% SDS in PBS, and three times with HEPES buffer (Hb: 50 mM HEPES pH 7.5, 100 mM NaCl, 4 mM $MgCl_2$, 0.2 mM TCEP added fresh). The beads were then treated as described below.

In Fig. 1C, the beads were treated with TssM* (2 µM) for 1 h at 37 °C. The supernatant was then separated from the beads and both fractions were quenched with 4× sample buffer (1×: 10% glycerol, 50 mM Tris-HCl (pH 6.8), 2% SDS, 1% β-mercaptoethanol, 0.02% bromophenol blue). The bead fraction was boiled at 95 °C for 5 min to elute GFP-PARP7 for detection by western blotting.

## Plasmids and cloning

DNA sequences for human RNF114, RNF138, and RNF166 were synthesized from Twist Biosciences and subsequently cloned into the pOPIN-B and pOPIN-S3C vectors (Berrow et al, 2007). Site-directed mutagenesis was used to insert substitutions into RNF114, and the products were confirmed with DNA sequencing.

## Protein expression

The following proteins were purified as previously reported: Human UBE1 (Gladkova et al, 2018), UBE2D3/DTX2 (Bejan et al, 2025), AF1521 (Kamata et al, 2019), TssM* (Szczesna et al, 2024), OTULIN (Keusekotten et al, 2013), LotA-N/Ub (Warren et al, 2023), Cezanne (Mevissen et al, 2016), and OTUB1*/AMSH* (Michel et al, 2015).

His-tagged RNF114 and RNF166 were expressed as N-terminal His-3C fusion proteins from the pOPIN-B vector and RNF138 was expressed as an N-terminal His-SUMO-3C fusion from pOPIN-S3C using fresh transformations of *E. coli* Rosetta cells. A single colony was used to inoculate a starter culture of selective Luria Broth (LB) (30 µg/mL kanamycin, 35 µg/mL chloramphenicol). The starter was used to inoculate larger LB cultures, which were grown at 37 °C until the $OD_{600}$ reached 0.4–0.6. In all, 50 µM $ZnCl_2$ was added just before induction with 0.2 mM isopropyl β-D-1-thiogalactopyranoside (IPTG) for overnight expression at 18 °C.

The cells were harvested by centrifugation at $4500 \times g$, and in the case of RNF114 and RNF166, pellets were resuspended in 25 mM Tris pH 7.4, 200 mM NaCl, 2 mM β-mercaptoethanol. Sigmafast protease inhibitor cocktail, PMSF, lysozyme, and DNase were added to the cell solutions prior to lysis by sonication. The lysate was clarified by centrifugation at $45,000 \times g$, at 4 °C for 40 min. The supernatant containing the His-tagged protein was applied to Cobalt resin, and the resin was washed extensively with 25 mM Tris pH 7.4, 200 mM NaCl, 2 mM β-mercaptoethanol. Bound proteins were eluted using the same buffer containing 250 mM imidazole. Eluted proteins were further purified using gel filtration chromatography on a HiLoad 16/600 Superdex75 in 25 mM Tris pH 7.4, 150 mM NaCl, 2 mM DTT. RNF138 was purified using the same protocol, with the exception that buffers were at pH 8.0. Fractions containing the protein of interest were pooled, concentrated, flash-frozen, and stored at −70 °C until use.

## Synthesis of clickable ADPr (Compound 8)

Methods and validation corresponding to the synthesis of our clickable ADPr analog can be found in the supporting information.

## Preparation of Ub-ADPr-TAMRA (Ub-ADPr$^T$)

Ub-ADPr was generated similar to previously described using DTX2 and our clickable ADPr analog (Bejan et al, 2025; Kelly et al, 2024; Zhu et al, 2024; Zhu et al, 2022). Briefly, a preparative amount of E2~Ub conjugate was formed using either WT Ub or Ub$^{K11R}$ and purified using a Superdex75 10/300 Increase column pre-equilibrated in 25 mM HEPES pH 7.4, 150 mM NaCl. This step is necessary to remove any remaining ATP or its hydrolyzed forms that have previously been shown to be alternative substrates of DTX2 (Zhu et al, 2022). Any remaining compound in the ubiquitylation reaction that contains an ADP-ribose would be ubiquitylated by DTX2, resulting in heterogenous products instead of homogenous Ub-ADPr. The purified conjugate was then mixed with 5 µM DTX2 and 0.2 mM compound **8** in 25 mM HEPES pH 7.4, 150 mM NaCl to enzymatically form Ub-ADPr. After 1 h at 37 °C, the reaction was diluted 1:10 in 25 mM NaOAc pH 4.5. Ub-ADPr was purified from other reaction components using cation exchange chromatography on a Resource S column over a gradient from 0 to 500 mM NaCl. Fractions corresponding to Ub-ADPr were pooled and concentrated to <150 µL and evaluated using intact mass spectrometry.

The fluorescent species, Ub-ADPr$^T$, was made using click chemistry between the ADPr (alkyne) of Ub-ADPr and the TAMRA molecule (azide). A 3× reaction mixture was made up using 1.5 mM Tris(3-Hydroxypropyltriazolylmethyl)amine (THPTA), 0.75 mM $CuSO_4 \cdot 5H_2O$, 0.3 mM TAMRA-azide, and 7.5 mM sodium ascorbate in PBS pH 7.4. This mixture was then diluted to 1× into the Ub-ADPr solution and mixed for 1.5 h at room temperature. Ub-ADPr$^T$ was separated from unconjugated TAMRA-azide using a HiLoad 16/600 Superdex75 size exclusion column pre-equilibrated in 25 mM NaOAc pH 4.5, 150 mM NaCl. Fractions corresponding to Ub-ADPr$^T$ were pooled and concentrated, and the concentration was measured using the TAMRA fluorophore. The stock of Ub-ADPr$^T$ was diluted back to a more neutral pH using 50 mM HEPES pH 7.4, 100 mM NaCl, prior to long-term storage at −70 °C.

## Gel-based ubiquitylation experiment

Ubiquitylation experiments were conducted at 37 °C with 0.2 μM E1, 5 μM UBE2D3, 50 μM Ub, 2 μM RNF114, 1 mM ATP, 5 mM MgCl$_2$, and 1 mM DTT in 50 mM HEPES, pH 7.4, and 100 mM NaCl. The reaction was quenched with sample buffer at the indicated timepoints and resolved on a 4/16.5% Tris-Tricine gel. The gel was stained with Coomassie blue to visualize ubiquitylation products.

## Fluorescence polarization experiments

Fluorescence polarization experiments were conducted using a BMG Labtech CLARIOstar microplate reader. Reaction volumes were 10 μL and data were collected in a black, low protein-binding 384-well plate at 22 °C. For all experiments, 25 mM HEPES pH 7.4, 150 mM NaCl, 0.1 mg/mL BSA, 1 mM TCEP (FP buffer) was used as the blank. The 540–20 nm and 590–20 nm optic filters were used to monitor the TAMRA fluorophore.

For binding experiments, a range of concentrations was chosen for each enzyme. Fresh dilutions were made in duplicate using FP buffer. Samples were mixed 1:1 using a 2× stock of the indicated fluorescent substrate (Ub-ADPr$^T$, ADPr$^T$, Ub-Lys$^T$) and protein. Ten measurements were recorded per sample, and an average was taken. To determine K$_d$ values of RNF114 with Ub-ADPr$^T$ or Ub$^{K11R}$-ADPr$^T$ and AF1521 with ADPr$^T$, Ub-ADPr$^T$, or Ub-Lys$^T$, the average baseline signal of the fluorescent substrate in the absence of binding partner was subtracted from each measurement, and final values were plotted as change in fluorescence polarization (ΔFP) against protein concentration. A nonlinear regression of the duplicate datasets was used in GraphPad Prism 10 to derive K$_d$ values with the ADPr$^T$, Ub-ADPr$^T$, Ub$^{K11R}$-ADPr$^T$ substrates.

A UbiReal experiment was used to monitor the ligase activity of proteins (Franklin and Pruneda, 2023). A reaction mixture containing 0.2 μM E1, 1 μM UBE2D3, 10 μM unlabeled Ub, 0.5 μM RNF114, 5 mM MgCl$_2$, with 50 nM of the indicated fluorescent substrate (Ub-ADPr$^T$, ADPr$^T$, Ub-Lys$^T$) was made up in FP buffer. For ligase reactions with RNF138 and RNF166, 5 μM ligase was used. A baseline FP was collected for 10 min, and then 1 mM ATP was added to the wells to initiate the ligase assay. The plate was put back into the plate reader and monitored for 70 more minutes. For reactions containing unlabeled ADPr, 0.2 mM was used.

For the UbiCRest experiment to diagnose the chain type formed by RNF114, the products of a UbiReal experiment were treated with TssM*, OTULIN, LotA, OTUB1*, or AMSH* at 0.5 μM concentration or with Cezanne at 0.02 μM concentration. The reaction was monitored by fluorescence polarization for the indicated amount of time. At the end of the reaction, samples were removed from the plate, quenched with sample buffer, and resolved on a 4–20% Mini-PROTEAN TGX gel (Bio-Rad Laboratories). Reaction products were visualized by in-gel fluorescence to track TAMRA-conjugated species.

## Thermal stability assay

A thermal stability assay for RNF114 mutants was conducted in MicroAmp Fast 96-Well Reaction Plates (Applied Biosystems) with SYPRO Orange Protein Gel Stain (MiliporeSigma) using a

QuantStudio 3 Real-Time PCR system (Applied Biosystems). Assays were performed in 20 μL volumes containing 10 μM RNF114 variants and 20× SYPRO dye (diluted from a 5000× stock) in 50 mM HEPES pH 7.4, 100 mM NaCl. The protocol increased the temperature from 22 to 99 °C on a gradient of 0.1 °C every 5 s, and fluorescence was monitored using an excitation wavelength of 580 ± 10 nm and an emission wavelength of 623 ± 14 nm.

## Mass spectrometry

Intact mass was determined using a ThermoScientific LTQ Velos Pro linear ion with a C18 trap. A Poroshell C18 column (1.0 Å × 75 mm × 5 μm) was used for reverse-phase high-performance liquid chromatography (RP-HPLC) prior to sample ionization. 0.5 μg of purified protein was resolved using a water/acetonitrile gradient from 2 to 50% acetonitrile in 0.1% formic acid. Data were acquired using Xcalibur (ThermoScientific) and processed using Freestyle (ThermoScientific) for spectrum deconvolution.

## Western blotting

Cells were lysed in cell lysis buffer (CLB; 50 mM HEPES pH 7.4, 150 mM NaCl, 1 mM MgCl2, 1% Triton X-100) supplemented with fresh 1 mM TCEP, 1× cOmplete EDTA-free Protease Inhibitor Cocktail (Sigma-Aldrich, 11873580001), 30 μM Phthal 01 (pan-PARP inhibitor) (Rodriguez et al, 2021), and 50 μM PR-619 (Medchemexpress, HY-13814). Lysates were centrifuged at 14,000 rpm for 10 min at 4 °C, quantified with a Bradford assay (Bio-Rad Laboratories, 5000006EDU), and resolved in 4–20% precast gels (Bio-Rad Laboratories, 4561096). Proteins were transferred to a nitrocellulose membrane using Trans-Blot Turbo Transfer System (Bio-Rad Laboratories), blocked in 5% milk (Carnation) in 1× PBS, 0.1% Tween-20 (PBST), and probed overnight at 4 °C with primary antibodies. After three washes with PBST, the blots were incubated for 1 h at room temperature with a goat anti-rabbit (1:10,000, Jackson ImmunoResearch Labs, 111035144) or goat anti-mouse (1:5000, Invitrogen, 62-6520) HRP-conjugated secondary antibody in 5% milk in PBST. After three washes with PBST, blots were developed with SuperSignal™ West Pico (Thermo Scientific, 34578) or Femto (Thermo Scientific, 34095) chemiluminescent substrate and imaged on a ChemiDoc Gel Imaging System (Bio-Rad Laboratories). The primary antibodies used in this study are: Mono-ADP-Ribose (AbD33204) (1:250; Bio-Rad Laboratories), GFP (pabg1) (1:2000, Proteintech), HA-Tag (C29F4) (1:1000; Cell Signaling Technologies), RNF114 (GTX107046) (1:1000; Genetex), DTX2 (PA5-60164) (1:500, Thermo Fisher Scientific). Western blots were quantified using ImageLab Software (BioRad).

## Reverse transcription quantitative PCR

RNA was extracted using the *NucleoSpin RNA Plus Kit* (Macherey-Nagel) following the manufacturer's protocol. Cells were lysed, filtered, and RNA was purified with on-column DNase treatment. RNA was eluted in RNase-free water and quantified using a NanoDrop. cDNA was made using the *ProtoScript II First Strand cDNA Synthesis Kit* (NEB). RNA was mixed with primers and

dNTPs, heated to 65 °C, then reverse transcribed at 42 °C for 1 h. The reaction was stopped at 80 °C. cDNA was stored at −20 °C. Real-Time PCR using the cDNA was performed using PowerUp SYBR green Master mix (ThermoFisher Scientific #A25742) and analyzed with a Quantstudio3 (Applied Biosystems). Relative quantities were determined by normalization to the housekeeping gene HRPT. Fold change was calculated using the delta-delta Ct method and the formula $2^{-\Delta\Delta Ct}$.

## Sequence and structural analysis

Structural modeling was performed using the AlphaFold3 web interface (Abramson et al, 2024). AlphaFold3 PAE plots were visualized using PAE Viewer (Elfmann and Stulke, 2023). ConSurf analysis of evolutionary conservation was performed using the ConSurf web interface (Ashkenazy et al, 2016; Ashkenazy et al, 2010; Celniker et al, 2013; Glaser et al, 2003; Landau et al, 2005). Structural analysis and figure generation were performed using PyMol (The PyMOL Molecular Graphics System, Version 3.0 Schrödinger, LLC). Sequence alignments were generated using T-Coffee and visualized with JalView (Waterhouse et al, 2009).

The evolutionary history presented in Fig. 5D was inferred using the UPGMA method (Sneath, 1973). The optimal tree with the sum of branch length = 2.106 is shown. The percentage of replicate trees in which the associated taxa clustered together in the bootstrap test (500 replicates) are shown next to the branches (Felsenstein, 1985). The evolutionary distances were computed using the Poisson correction method (Zuckerkandl and Pauling, 1965) and are in units of the number of amino acid substitutions per site. The analytical procedure encompassed 4 amino acid sequences. The pairwise deletion option was applied to all ambiguous positions for each sequence pair, resulting in a final dataset comprising 500 positions. Evolutionary analyses were conducted in MEGA12 (Kumar et al, 2024; Stecher et al, 2020) utilizing up to seven parallel computing threads.

# Data availability

This study includes no data deposited in external repositories. Reagents will be made available upon request to the corresponding authors.

The source data of this paper are collected in the following database record: biostudies:S-SCDT-10_1038-S44318-025-00577-z.

# Peer review information

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

## Acknowledgements

The authors wish to thank all members of the Pruneda and Cohen labs for advice and discussion contributing to the experimental design and data analysis for this work. Mass spectrometric analysis was performed within the OHSU Proteomics Shared Resource with partial support from NIH core grants P30EY010572, P30CA069533, and S10RR025571. This work was supported by the National Institute of Neurological Disorders and Stroke (2R01NS088629 to MSC) and by the National Institute of General Medical Sciences (1T32GM142619 to JKS and R35GM142486 to JNP).

## Author contributions

**Rachel E Lacoursiere**: Conceptualization; Investigation; Visualization; Methodology; Writing—original draft; Writing—review and editing. **Kapil Upadhyaya**: Conceptualization; Investigation; Methodology; Writing—review and editing. **Jasleen Kaur Sidhu**: Investigation; Writing—review and editing. **Ivan Rodriguez Siordia**: Investigation; Writing—review and editing. **Daniel S Bejan**: Investigation; Writing—review and editing. **Michael S Cohen**: Conceptualization; Supervision; Funding acquisition; Writing—review and editing. **Jonathan N Pruneda**: Conceptualization; Supervision; Funding acquisition; Writing—review and editing.

Source data underlying figure panels in this paper may have individual authorship assigned. Where available, figure panel/source data authorship is listed in the following database record: biostudies:S-SCDT-10_1038-S44318-025-00577-z.

## Disclosure and competing interests statement

The authors declare no competing interests.

# Expanded View Figures

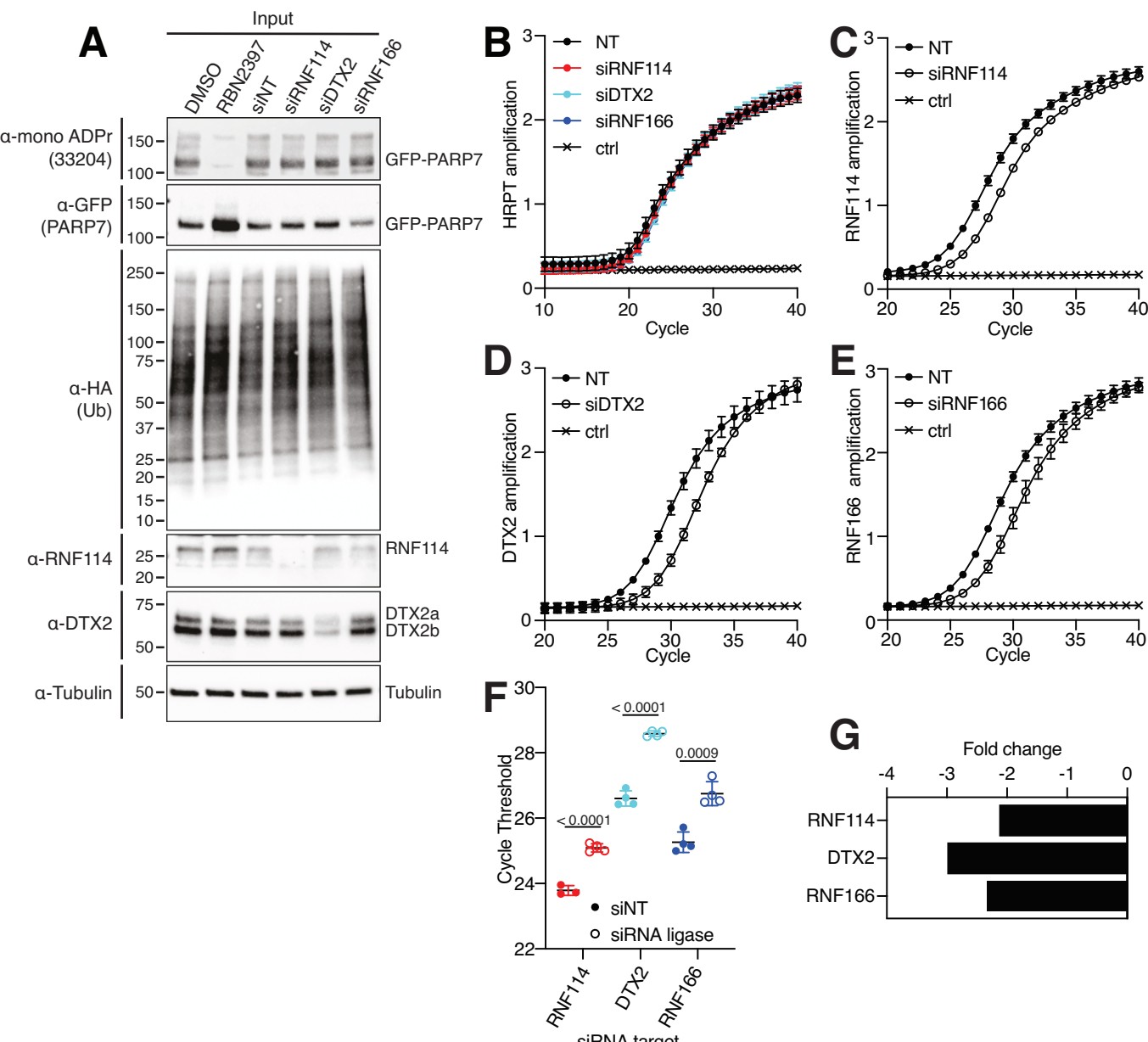

**Figure EV1.  RNF114 extends PARP7 MARUbylation.**

(A) Input blots corresponding to Fig. 1C prior to the MARUbylation assay to separate PARP7-bound canonical ubiquitylation from MARUbylation. Genetic knockdown was further confirmed using reverse transcription quantitative PCR (RT-qPCR) and the amplification curves are shown for (B) housekeeping gene HRPT, (C) RNF114, (D) DTX2, and (E) RNF166. In these panels, the ctrl refers to a no cDNA control generated from the siNT transfected cells. Data show the mean ± standard deviation for $n = 3$ technical replicates. (F) The cycle threshold for amplification of the indicated ligases are shown for the siNT treatment and the respective siE3 treatment. The mean ± standard deviation cycle threshold for $n = 4$ technical replicates for each siE3 treatment was compared to that for the siNT treatment. Significance was assessed using an unpaired $t$ test and $P$ values are shown on the graph where ***$0.001 > P > 0.0001$, ****$P < 0.0001$. (G) Fold change in gene expression was calculated using the delta-delta Ct method.

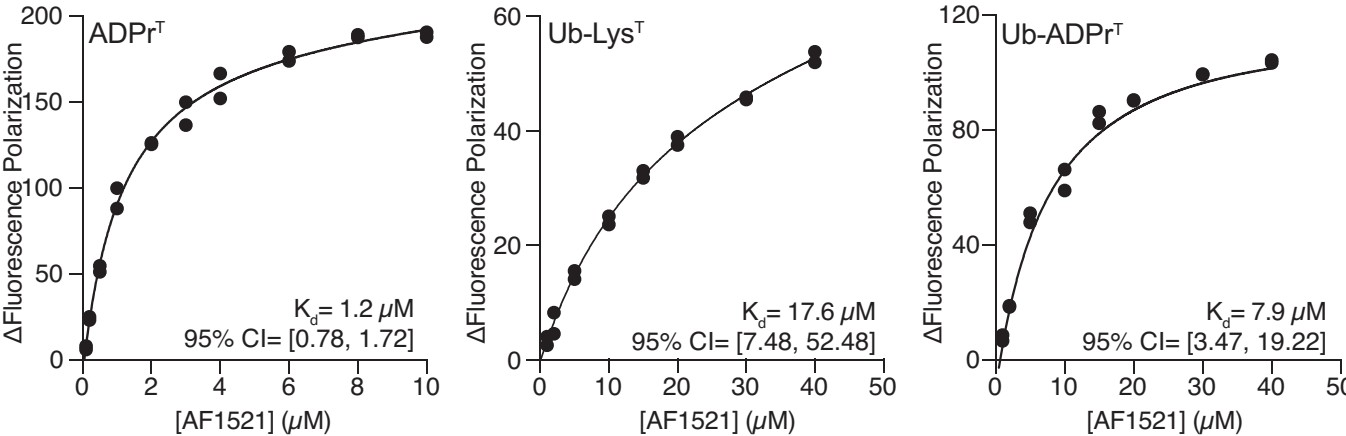

**Figure EV2. Validation of fluorescent ADPr$^T$.**

Fluorescence polarization binding experiment of ADPr$^T$, Ub-Lys$^T$, and Ub-ADPr$^T$ with the AF1521 macrodomain to validate the substrates against a known ADPr-binding protein. $K_d$ values and the associated 95% confidence intervals (CI) were derived using GraphPad Prism 10 using a one-site total binding fit.

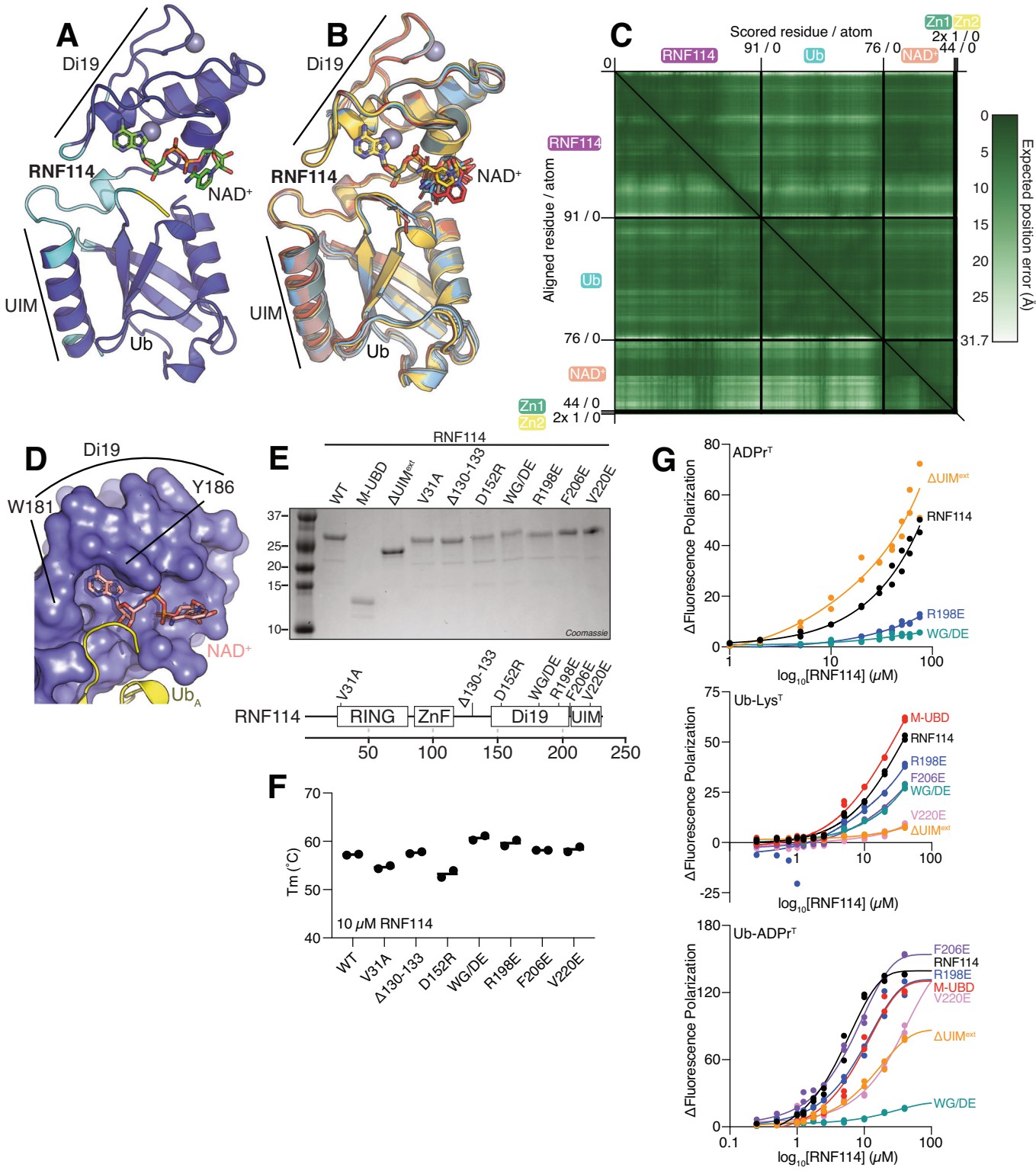

**Figure EV3. RNF114 binds NAD$^+$ and Ub.**

(A) The M-UBD of RNF114 was modeled in AlphaFold3 with two Zn$^{2+}$ ions, NAD$^+$, and Ub. The cartoon diagram of the complex is shown with NAD$^+$ in green and protein colored by AlphaFold3 confidence (pLDDT) where blue represents pLDDT > 90, cyan 70 > pLDDT > 90, yellow 50 > pLDDT > 70, and orange pLDDT < 50. The Di19 domain and UIM that make up the M-UBD of RNF114 are labeled. (B) Superimposition of the top 5 AlphaFold3 models of the M-UBD:NAD$^+$:Ub complex in different colors. (C) PAE plot from AlphaFold3 of the M-UBD:NAD$^+$:Ub model shown in (A). Darker green indicates higher confidence in the model due to a lower expected position error. (D) W181 and Y186 contribute to the positioning of NAD$^+$ in the binding pocket. Surface representation of the RNF114 Di19 domain accentuates the pocket that accommodates the adenine ring of NAD$^+$. (E) RNF114 mutants used in this study were diluted to 1 µM and visualized by SDS-PAGE and Coomassie staining. A schematic of RNF114 and the location of the chosen mutations is also shown. (F) Melting temperatures of RNF114 and its mutants derived from a thermal stability assay. (G) Fluorescence polarization binding curves for the indicated RNF114 mutants with either ADPr$^T$, Ub-Lys$^T$, or Ub-ADPr$^T$ as shown in Fig. 3E instead showing the RNF114 concentration (x-axis) on a log scale for ease of comparison.

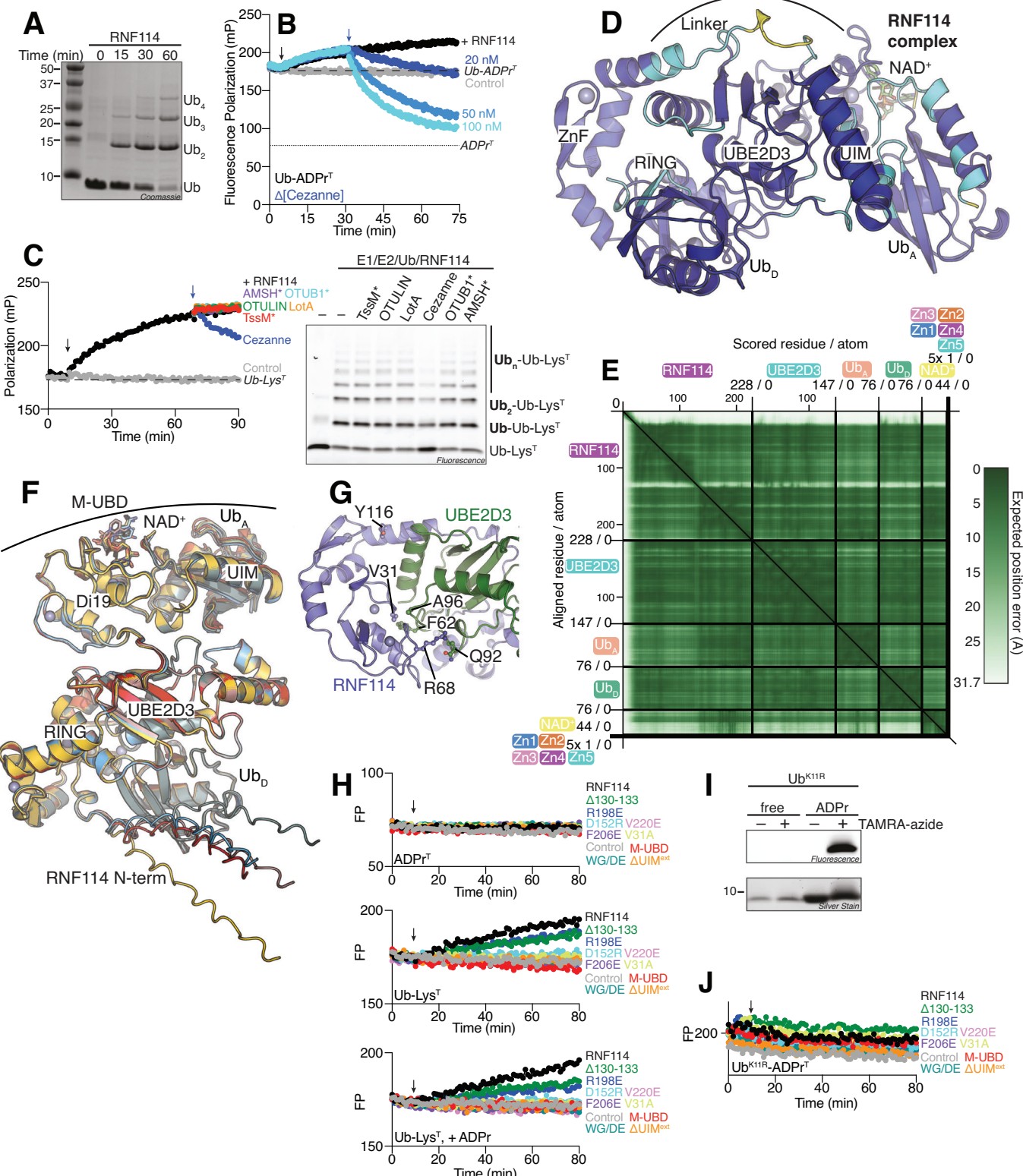

**Figure EV4.  RNF114 catalyzes K11-linked extension on Ub-Lys.**

(A) RNF114 autoubiquitylation with WT Ub. Samples were removed at the indicated timepoints and quenched in 3X sample buffer prior to separation by SDS-PAGE and Coomassie staining. (B) UbiCRest experiment optimizing the concentration of Cezanne to target the K11 isopeptide linkage preferentially over the Ub-ADPr ester linkage. This experiment was conducted as described in the Materials and Methods with the indicated concentrations of Cezanne to identify 20 nM as the optimal Cezanne concentration for our experiments. (C) UbiCRest experiment using the indicated deubiquitylases for RNF114 ubiquitylation of Ub-Lys$^T$. The reaction was monitored by fluorescence polarization and the sample was removed from the plate at 90 min, run on SDS-PAGE, and visualized by in-gel fluorescence of the TAMRA fluorophore. (D) AlphaFold3 model of the full-length RNF114, NAD$^+$, UBE2D3, two copies of Ub, and five Zn$^{2+}$ ions. The cartoon diagram of the complex is shown with NAD$^+$ in green and protein colored by AlphaFold3 confidence (pLDDT) where blue represents pLDDT > 90, cyan 70 > pLDDT > 90, yellow 50 > pLDDT > 70, and orange pLDDT <50. The RING and ZnF domains and UIM of RNF114, UBE2D3, Ub$_D$, the UIM-bound Ub (Ub$_A$), in addition to the linker of RNF114 that mediates backside UBE2D3 binding are labeled. (E) PAE plot from AlphaFold3 for the RNF114 transferase complex shown in (D). Darker green indicates higher confidence in the model due to a lower expected position error. (F) The top 5 AlphaFold3 models from (D) are shown aligned and colored differently. Each protein and specific regions of RNF114 are labeled. (G) Detailed view of the modeled interface between the RNF114 RING domain and UBE2D3. V31 and the linchpin residue R68 are shown from the RING domain of RNF114 and their proximity to F62/A96 and Q92, respectively. Y116, a tyrosine phosphorylation site, is also shown in this view. (H) UbiReal curves showing the activity of RNF114 mutants against the ADPr$^T$ and Ub-Lys$^T$ ± unlabeled ADPr substrates. These graphs show a representative dataset of $n = 2$ technical replicates. (I) Validation of the Ub$^{K11R}$-ADPr$^T$ substrate (products of the CuAAC reaction) by SDS-PAGE and in-gel fluorescence. (J) UbiReal experiment for RNF114 mutants against the Ub$^{K11R}$-ADPr$^T$ substrate. For all fluorescence polarization panels, the black arrow represents the timepoint when ATP was added to the reaction to initiate RNF114 ligase activity, and the blue arrow represents the timepoint when the indicated deubiquitylase was added to cleave the products.

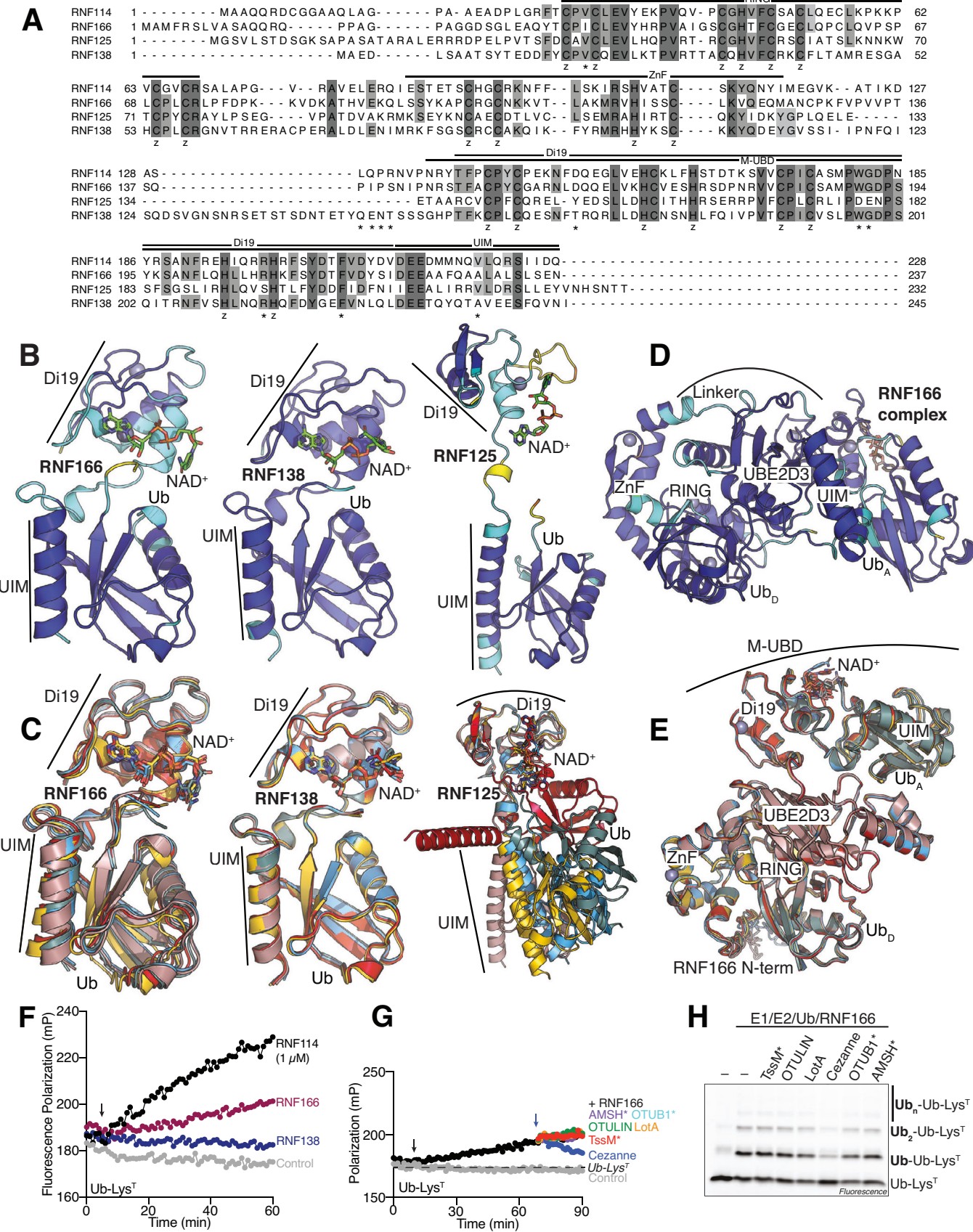

◀ **Figure EV5. Ub-ADPr recognition is conserved in a family of MARUbe-Targeted Ligases (M-UTLs).**

(A) Sequence alignment of M-UTLs RNF114, RNF166, RNF125, and RNF138. The RING and ZnF domains, M-UBD, Di19, and UIM are labeled based on the sequence of RNF114. Sites of point mutation in our experiments with RNF114 are indicated by asterisks, and $Zn^{2+}$-coordinating residues are labeled (z). Sequences were aligned using Jalview and colored according to percent identity where the darker the gray corresponds to higher conservation. (B) Cartoon diagrams of the M-UBD of RNF166, RNF138, or RNF125 in complex with Ub and $NAD^+$ colored by the AlphaFold3 confidence scale. (C) Alignment of the top 5 models for each M-UBD (RNF166, RNF138, or RNF125) in complex with $NAD^+$ and Ub. (D) AlphaFold3 model of RNF166 in complex with UBE2D3, $NAD^+$, two copies of Ub, and five $Zn^{2+}$ ions, colored by confidence. (E) Overlay of the top 5 AlphaFold3 models for the RNF166 ligase complex. (F) Representative UbiReal experiment for the indicated E3 ligases with the Ub-Lys$^T$ substrate. In this experiment, RNF114 was used at 1 µM and the other ligases were used at 5 µM. ATP addition is marked by the black arrow. (G) Reaction products of a ligase assay for RNF166 with Ub-Lys$^T$ were subjected to a UbiCRest panel. ATP was added to initiate RNF166 ligase activity at the black arrow, and the indicated deubiquitylase was added at the blue arrow. (H) The 90-min timepoint from (G) was removed from the plate reader and visualized by SDS-PAGE and in-gel fluorescence. (B, D) For all AlphaFold3 confidence coloring, blue represents pLDDT >90, cyan 70 > pLDDT >90, yellow 50 > pLDDT >70, and orange pLDDT <50.

