## [Peer Review File · The EMBO Journal]

RNF114 and RNF166 exemplify reader-writer E3 ligases that extend K11 polyubiquitin onto sites of MARUbylation

Rachel Lacoursiere, Kapil Upadhyaya, Jasleen Kaur Sidhu, Ivan Rodriguez Siordia, Daniel Bejan, Michael Cohen, and Jonathan Pruneda

Corresponding author(s): Jonathan Pruneda (pruneda@ohsu.edu) , Michael Cohen (cohenmic@ohsu.edu)

Review Timeline:

Submission Date:	15th May 25
Editorial Decision:	8th Jun 25
Revision Received:	7th Aug 25
Editorial Decision:	1st Sep 25
Revision Received:	5th Sep 25
Accepted:	16th Sep 25

Editor: Hartmut Vodermaier

Transaction Report:

Dr. Jonathan N Pruneda
Oregon Health & Science University
Molecular Microbiology & Immunology
3181 SW Sam Jackson Park Rd
Mail Code L220
Portland, OR 97239

8th Jun 2025

Re: EMBOJ-2025-121383
A family of E3 ligases extend K11 polyubiquitin on sites of MARUbylation

Dear Jonathan,

Thank you again for submitting your study on ubiquitin ligases recognizing and modifying MARUbylation sites. We have now received a complete set of comments from three referees with expertise in ADR-ribosylation, E3 ligases, and structural biology, copied below for your information. Since all referees are in principle supportive of the study, we would be happy to consider it further for EMBO Journal publication, pending adequate revision of a number of specific technical and presentational issues raised by the reviewers. These include controls for mutant protein functionality, questions about protein detection on blots, knockdown validation, as well as decisiveness of some of the data and possible alternative interpretations; and other points that I will not reiterate in detail here.

Please remember that it is our policy to consider only a single round of major revision, making it important to fully respond to all comments at the time of resubmission; therefore, please do not hesitate to get back to me in case you would like to clarify/discuss any of the referees' points or plans for addressing them ahead of time. We would also be open to extending the revision deadline if that should be helpful. As always, our scooping protection will remain valid throughout the full revision period.

Detailed information on preparing, formatting and uploading a revised manuscript can be found below and in our Guide to Authors, and adhering to them as closely as possible shall greatly facilitate editorial processing upon resubmission. Thank you again for the opportunity to consider this work for The EMBO Journal, and I look forward to your revision in due time.

With kind regards,

Hartmut

9) To facilitate reproducibility and cross-laboratory adoption of methodologies, please structure the Materials & Methods section as outlined in our guide to authors, including a completed Reagents and Tools Table that can be downloaded from our author guidelines as well (<https://www.embopress.org/page/journal/14602075/authorguide#structuredmethods>).

10) Digital image enhancement is acceptable practice, as long as it accurately represents the original data and conforms to community standards. If a figure has been subjected to significant electronic manipulation, this must be clearly noted in the figure legend and/or the 'Materials and Methods' section. The editors reserve the right to request original versions of figures and the original images that were used to assemble the figure. Finally, we generally encourage uploading of numerical as well as gel/blot image source data; for details see: embopress.org/page/journal/14602075/authorguide#sourcedata

In the interest of ensuring the conceptual advance provided by the work, we recommend submitting a revision within 3 months (6th Sep 2025). Please discuss the revision progress ahead of this time with the editor if you require more time to complete the revisions. Use the link below to submit your revision:

Link Not Available

Referee #1:

General

The presented work from the Pruneda and Cohen labs follows up where their last study left: in previous work, they showed that in principle, MARUbylation can occur in human cells. In the current work, they find that PARP7 is MARUbylated in cells on glutamate/aspartate and continue with synthesis of ubiquitin-ADPr-TAMRA. This was then used to study a protein, RNF114, which they could show specifically binds MARUbe and extends the modification with K11-linked polyubiquitin. Lastly, they test several other RNF proteins (containing a Di19 and UIM) as potential MARUbe binders. In general, this work is of high interest as it adds another piece to the ADPr-ubiquitin puzzle by adding readers specific for MARUbe. There are some issues which need to be resolved to further strengthen the key findings.

Major

The use of a C5 mutant is not very convincing. Do the authors have evidence that the protein folds normally and retains catalytic activity? In fact, the mutant is perhaps not necessary at all: neutral hydroxylamine will not remove the modification from cysteine. Performing this experiment with the wildtype PARP7 and seeing release of MARUbe with neutral hydroxylamine already indicates that either glu/asp or arg are modified. The authors should indicate how long samples were incubated with neutral hydroxylamine, as this is important to distinguish between glu/asp (fast release) and arg (slow release).

Even though I am aware that western blots are semiquantitative at best, it might be worthwhile quantifying the reduction in MARUbylation following RNF114/DTX2 knockdown. The authors describe these enzymes as the major enzymes responsible for the modification, but it looks like a significant MARUbe signal is still present. Do the authors think that other enzymes might have similar functions (other DTX/RNF/?), or is the knockdown not efficient enough to get larger effects?

I appreciate the authors' honesty in displaying blots where knockdown cannot be verified (supp figure 1). It would be good to find a way to estimate the levels of protein remaining if the DTX2 and RNF166 antibodies are not suitable. Perhaps using different antibodies, or if knockdown cannot be verified on protein level a qPCR could perhaps be done to measure downregulation on RNA level? If this is not possible, I would at the very least recommend to mention in the text/figure legends that knockdown levels could not be determined.

The images in Figure 3 can be described better. If the models and data are correct, RNF114 binds to a hybrid ADPr-ubiquitin. From the text and figure legend I understand that both NAD and ubiquitin are modelled, but in the figure NAD and MARUbe are labeled. I find this slightly misleading, as the "MARUbe" is then in fact ubiquitin. Would it be possible to perform the modeling with ADPr and ubiquitin instead of NAD, or even generate a ADPr-ubiquitin to model? I would be interested to see where the ubiquitin is attached to ADPr in this model. The fact that the nicotinamide was present in several orientations (Supp Fig 4F/5C) probably simply reflects the fact that when binding ADPr-Ub, RNF114 does normally not interact with nicotinamide.

Furthermore, several RNF114 mutants are tested (Fig 3E), which in principle is a valid strategy to determine how those residues contribute to binding of MARUbe. I cannot find verification of the mutant proteins: are they folding properly? Could their function as ubiquitin E3 ligases be tested to verify folding? At the very least, the authors should include Coomassie stained gels of the proteins used, to provide some basic information about the proteins used.

Minor

The mentioned analog sensitive chemical genetics is indicated as unpublished data (used for figure 1B). Will those data be included with the manuscript, or do the authors plan to publish this dataset at a later timepoint?

The authors discuss the K11-linked ubiquitin in Figure 1D, but it wasn't formally shown that this indeed is K11-linked. This can probably be confirmed by a K11-linkage antibody, DUB treatment or otherwise differently phrased - in later figures the authors do test the nature of linkages using DUBs.

The authors should perhaps reconsider their suggested naming "MUBD" and "MUTL". The DNA repair protein "MutL" exists in E.coli and using the same abbreviation might lead to confusion, even though the human version is named "MutL protein homolog 1". MUBD is not used as abbreviation yet as far as I know, but may be hard to pronounce when presenting these data?

Figure 4B/C, the figure legend is not quite clear. "After 30 minutes, the plate was removed from the instrument and samples were quenched in SDS sample buffer". Does this mean that the sample visualised in Figure 4C was taken at t=120? Also, in the text the authors describe that a {plus minus}50% reduction in signal was observed, but it is not clear how this value was calculated?

It would be good to indicate amounts of enzymes used in the figure legends. It is for example indicated in the text that Cezanne was used at 20nM in some experiments, but I'd recommend to include this type of info with the figures as well.

The authors might want to include a little paragraph on how RNF166 functions in cells, considering that they do not see any effects of RNF166 knockdown in 1D? Might be worthwhile to expand the discussion to include some information about RNF166.

Referee #2:

Reviewer comments

The authors describe Ubiquitin E3 ligases that recognize specifically recently discovered dual post-translational modification where Ubiquitin is attached to mono-ADP-ribosyl groups added to proteins by PARP enzymes. The topic of the study is very relevant and timely and there are multiple studies ongoing and available as preprints currently as the authors also point out in the manuscript. The manuscript is well structured and has a logical progression between the experiments. The claims are mostly supported by the data provided, but there are a few major concerns that should be addressed before publication.

Major concerns

Already in the title the authors talk about a family of E3 ligases but in the end, it is just two enzymes, RNF114 and RNF166, that have the described activity. The title should reflect the actual data better in this respect and be more specific.

While the figures are visually nice, the authors should help the reader and label the figures better and also point out things to

look at in the figure in the main text to improve readability. As examples figure 1A supernatant bands should be labelled like in Figure 4C and MARUbe is not MARUbe in the models but just Ub (like defined in the figure legends).

In Figure 1A, GFP WB detection exhibits a single band, as opposed to the Ub and MAR detection from which a smear is seen. Why is this? One could expect the signal to be low or even undetectable but not change behaviour. It is only in the input panel of Figure 1A where a smear is noticeable also in with the GFP antibody. It would be good to provide the full blots in the supplement.

In the context of Figure 1D, the authors provide verification of the knockdown in the supplementary information. This is, however, not convincing. It appears that DTX2 antibody is not working at all specifically (full blot not provided) and that siRNF166 is not functional. This puts into question the conclusions made that RNF114 would be the main E3 ligase as the RNF166 is always present in the experiments. The nonfunctional knockdown agrees well with the data presented. The claim that the blot had a high background is not justified in these cases that are some of the key results of the manuscript.

The binding of RNF138 to UB-ADPrT is not convincing. Taken that RNFs are the about the same MW one would expect the anisotropy signal to be at the same level in Figure 5E. The observed binding appears to be unspecific. This is also in line with Figure 5F that shows RNF138 being inactive.

Minor points and corrections

The language is good, but one could consider some adjectives to be removed to make it a bit more scientific (e.g. "strikingly", "newly").

Please use TNKS1 and TNKS2 as acronyms for tankyrases and if the point is to mention that these belong to the same family add "formerly called PARP5a/b"

In the Figure 1D the HA-Ub identification for the siDTX2 treatment seems to be in agreement with the claim that DTX2 plays a major role in the formation of the initial MARUbe as the 10 kDa band is mostly gone, but one would expect that this would also cause that the 20-25 kDa bands would have also had a considerable lower signal as there would be no place for them to dock. This is not the case, and the signal is similar to siRNF114 treatment. Is this related to another ligase that would be compensating or nonfunctional knockdown?

The last experimental section is a nice combination of evolutionary conservation and biochemistry to elucidate the activity in vitro of two of the newly described enzymes. However, given that in the discussion the poor conservation of RNF125 is mentioned again, would it have been possible to make a reconstruction of the evolutionary divergence of the enzymes? This would shed light on whether RNF125 is "evolving towards a different substrate" or if on the other hand, MARUbylation is a more recently developed system.

Page 5, paragraph 1, line 7: remove phrases such as "a bit more complicated" for economy of language and reduce overall ambiguity.

Page 5, paragraph 2, it would help to clarify in the text which lanes of the WB are being compared.

Figure 1B TssM cleavage site missing in the schematic.

Figure 1B, necessary to add the second Ub to make a visual clarification of the reason behind the two bands in the supernatant fractions?

In Figure 1C please add the references to the studies to the figure legend. One cannot now evaluate one of the studies that is just referred to as "data unpublished". Would it be possible to add to be published elsewhere?

One should change "PARP7-specific inhibitor RBN2397" to "PARP7 inhibitor RBN2397" based on the supplementary information of the original publication.

Page 7, paragraph 2, line 10-11, not necessary to direct the reader to MM?

Page 10, paragraph 1 (continuation of the last paragraph of Page 9), line 1, hereinafter would be more accurate.

Page 14, paragraph 1 (continuation of last paragraph of page 13), line 7-8, refer to Perrard and Smith (2023) in which they also show that the UIM is necessary for the efficient modification of TNKS by RNF166.

Figure 5 B,C, the NAD and MARUbe (Ub as suggested in a previous point), could be re-arranged so as not to block the figure.

Overall, one could change the yellow label in the MARUbe for a more visible colour.

Having used structural predictions extensively, it would have been nice to see a full transfer complex with RNF138 that could perhaps indicate if indeed the linker length is crucial for the MUTLs to orientate ubiquitin correctly for the reaction to occur. One could also experiment by changing the linker length, but authors may want to reinspect the conservation surface also due to the lack of activity and that appears quite different from RNF166.

It would be appropriate to add standard deviations for the measured dissociation constants.

Referee #3:

The manuscript by Lacoursiere et al (from the laboratories of Michael Cohen and Jonathon Pruneda) focuses on deciphering how K11-linked ubiquitin chains are assembled on proteins that have been previously modified with ADP-ribose, that has then been ubiquitinated i.e. the target for K11 chain assembly already has an ADPr-Ub modification.

This study builds on prior work by these groups (Bejan et al., EMBO J 2025) and that of others. There is also a similar study available as a preprint <https://www.biorxiv.org/content/10.1101/2025.05.08.652854v1>

The key discovery described is the identification of RNF114 as an E3 ligase that binds ADP-ribose linked to Ubiquitin via two zinc binding domains and the C-terminal UIM (ubiquitin interaction motif). The development of labelled ADPr-Ub is critical to the approach taken and this reagent will likely enable other ADPr-Ub binding proteins to be characterised.

Having identified RNF114, AlphaFold is used to prepare models of the complexes that are then supported by mutagenesis. Following binding to ADPr-Ub, RNF114 appears to then specifically promote the attachment of another ubiquitin molecule to Lys11 of the ADPr-Ub moiety. This is a remarkable example of modifying enzymes building considerable post-translational complexity on proteins. Following characterisation of RNF114 they then analyse other related E3 ligases and discover that RNF166 has similar properties to those of RNF114, but that the RNF138 and RNF125 - the other two related proteins differ.

The manuscript contains considerable data and will be of interest to those in the field, but as written it is not necessarily easily accessible to a wider audience. The modifications are complex and as a result care is required to allow figures to be readily interpreted - some general simple schematics might help. In addition, abbreviations should be reviewed - not all are helpful. In addition the authors may wish to consider the order of the text as in some cases reordering of paragraphs might more readily lead the reader through the figures.

Main points:

- 1) The Abstract could be simplified to better articulate the key question addressed and the findings to make the research more readily available to a wider audience.
- 2) Figure 1A is complex and not well described. Close review of Bejan et al., EMBO J 2025 is required to interpret the figure. In addition, many parts of the figure are not described until later and the initial point made in paragraph 2 of the results takes some discerning for the non-specialist reader. It would be helpful to have another simple panel that introduces the reader to the modification before introducing the more complex experiment that follows from panel 1B. Alternatively the authors might consider reordering parts of the text and numbering the lanes so that specific lanes can be highlighted.
- 3) Figure 1D is critical for the focus on RNF114, and while the data clearly suggests RNF114 is important, it does suggest that others E3s likely also have important roles as ~50% of modification remains following knockdown of RNF114. It is also interesting that the authors conclude RNF166 is not important for modification even though Supp Fig1 does not show evidence of knockdown - it seems an alternate conclusion could be that RNF166 was not reduced? The conclusions should be more carefully articulated with the caveats highlighted.
- 4) The AlphaFold models of the predicted complex with ADPr-Ub are compelling and the associated mutagenesis data is nice. The authors note that while F206 is predicted to make a key contact, the F206E mutation has little impact on banding. In the model F201 appear to be located nearby - is it possible that F201 occupies the position where F206 is modelled?
- 5) In Figure 4 the E3 ligase activity is investigated and a model of the complex that accounts for the K11-linkage specificity proposed. This appears to nicely account for the assembly of K11-linked diUB but it is not clear how a K11-linked chain would be specified. The authors imply that longer chains can be built that have K11 links so it would be good to account for this or indicate uncertainty if the assembly of long chains is difficult to explain.
- 6) The data presented in Figure 4C is not as compelling as the related text. It might be helpful to highlight where the differences are and better explain the limitations of the data.
- 7) Throughout Di19 is used to refer to the commonly described ZnF2 and ZnF3 domains in RNF114, and there is a proposal to name Di19 plus the UIM as MUBD. The term Di19 is not widely used and is somewhat a naming anomaly, and it is not clear that the term MUBD would bring clarity to the field. Furthermore the authors indicate that RNF125 is unlikely to bind ADPr-Ub, and

RNF138 does not ubiquitylate Ub-ADPr, yet they propose to name the family of four proteins that includes RNF125 and RNF138 as MARUbe-targeted ligases (MUTLs). It is not clear that this is a helpful proposal.

Minor points:

- 1) The first paragraph of the results is quite confusing and in part refers to the previous paper, but uses the phrase 'first we sought to'.
- 2) Figure 2 describes preparation of labelled ADPr-Ub that is a key tool that underpins subsequent experiments. It might be helpful to number the steps in Figure 2A and refer to each in turn in the text. This would help the reader.
- 3) It would be helpful to include a schematic of RNF114 with the location of all the mutations highlighted.
- 4) In Figure 4A the preferential modification of ADPr-Ub is shown and the authors conclude that the ADPr-Ub linkage is recognised - it is also possible that it simply binds the linked substrate more tightly and not the linkage is recognised. This could be made clearer.
- 5) The scale used for the PAE plot should be indicated.
- 6) Throughout, but particularly in Figure 5 it would be helpful to have more whitespace between the panels in the Figure.

EMBOJ-2025-121383

We would like to thank all of the reviewers for their time and valuable feedback. As described in detail below, we have carefully considered and addressed each of the reviewer comments in the revised manuscript. Changes in the main text are highlighted in yellow.

Referee #1:

General

The presented work from the Pruneda and Cohen labs follows up where their last study left: in previous work, they showed that in principle, MARUbylation can occur in human cells. In the current work, they find that PARP7 is MARUbylated in cells on glutamate/aspartate and continue with synthesis of ubiquitin-ADPr-TAMRA. This was then used to study a protein, RNF114, which they could show specifically binds MARUbe and extends the modification with K11-linked polyubiquitin. Lastly, they test several other RNF proteins (containing a Di19 and UIM) as potential MARUbe binders. In general, this work is of high interest as it adds another piece to the ADPr-ubiquitin puzzle by adding readers specific for MARUbe. There are some issues which need to be resolved to further strengthen the key findings.

We thank the reviewer for their comments and constructive feedback.

Major

The use of a C5 mutant is not very convincing. Do the authors have evidence that the protein folds normally and retains catalytic activity? In fact, the mutant is perhaps not necessary at all: neutral hydroxylamine will not remove the modification from cysteine. Performing this experiment with the wildtype PARP7 and seeing release of MARUbe with neutral hydroxylamine already indicates that either glu/asp or arg are modified. The authors should indicate how long samples were incubated with neutral hydroxylamine, as this is important to distinguish between glu/asp (fast release) and arg (slow release).

The reviewer raises a fair point. Although our unpublished work shows that mutations of the major MARylated cysteines do not affect PARP7-mediated trans-MARylation (based on *in vitro* studies), more research is needed to fully understand the functional effects of mutating these cysteines. To precisely identify the amino acid-ADPr conjugates that serve as MARUbylation sites, we believe chemical treatments alone are insufficient and that optimized MS/MS-based approaches are essential for comprehensive characterization. Thus, we have removed the C5 mutant data from our manuscript and any discussion on site-specific MARUbylation, which will be the subject of a future study.

Even though I am aware that western blots are semiquantitative at best, it might be worthwhile quantifying the reduction in MARUbylation following RNF114/DTX2 knockdown. The authors describe these enzymes as the major enzymes responsible for the modification, but it looks like a significant MARUbe signal is still present. Do the authors think that other enzymes might have similar functions (other DTX/RNF/?), or is the knockdown not efficient enough to get larger effects?

We have completed additional repeats of this experiment and quantified the MARUbylation signals following siRNA treatments. The data show a statistically significant contribution of DTX2 and RNF114 toward the formation of K11-extended MARUbylation. In both cases we observe only a partial reduction, which could be a result of incomplete siRNA knockdown or functional redundancy among Deltex E3s and/or M-UTLs. We elaborate on this point further in the revised text.

I appreciate the authors' honesty in displaying blots where knockdown cannot be verified (supp figure 1). It would be good to find a way to estimate the levels of protein remaining if the DTX2 and RNF166 antibodies are not suitable. Perhaps using different antibodies, or if knockdown cannot be verified on protein level a qPCR could perhaps be done to measure downregulation on RNA level? If this is not possible, I would at the very least recommend to mention in the text/figure legends that knockdown levels could not be determined.

We have now confirmed knockdown by western blot for RNF114 and DTX2 and used RT-qPCR to demonstrate that RNF166 was also knocked down to a similar extent as RNF114 and DTX2.

The images in Figure 3 can be described better. If the models and data are correct, RNF114 binds to a hybrid ADPr-ubiquitin. From the text and figure legend I understand that both NAD and ubiquitin are modelled, but in the figure NAD and MARUbe are labeled. I find this slightly misleading, as the "MARUbe" is then in fact ubiquitin. Would it be possible to perform the modeling with ADPr and ubiquitin instead of NAD, or even generate a ADPr-ubiquitin to model? I would be interested to see where the ubiquitin is attached to ADPr in this model. The fact that the nicotinamide was present in several orientations (Supp Fig 4F/5C) probably simply reflects the fact that when binding ADPr-Ub, RNF114 does normally not interact with nicotinamide.

We apologize for the confusion. This ubiquitin was labeled MARUbe for consistency in helping distinguish the two ubiquitin molecules in later modeled complexes where an E2 is also present. We have relabeled MARUbe as Ub_A to denote its role as a K11 linkage acceptor. We chose to model the complex with NAD as a proxy for ADPr because only NAD is an available molecule within the publicly available AlphaFold3 package. NAD nicely mimics ADPr in this case, as the Ub C-terminus is nicely within reach of the 3' OH that is known to be modified by DTX2 (Zhu et al. 2022, Sci Adv) and the nicotinamide is oriented outside of the binding pocket. We highlight the conformational heterogeneity of the nicotinamide group among AlphaFold3 models as a reassurance that the models are consistent with binding to ADPr or ADP-ribosylated proteins.

Furthermore, several RNF114 mutants are tested (Fig 3E), which in principle is a valid strategy to determine how those residues contribute to binding of MARUbe. I cannot find verification of the mutant proteins: are they folding properly? Could their function as ubiquitin E3 ligases be tested to verify folding? At the very least, the authors should include Coomassie stained gels of the proteins used, to provide some basic information about the proteins used.

This is an excellent point. All mutants of RNF114 were purified alongside the wildtype protein using the same methodology. There were no inconsistencies in the elution profiles of each

mutant from size exclusion chromatography. We have moved our Coomassie-stained gel of RNF114 mutants from Supplemental Figure 4H to Supplemental Figure 3F, so that it can be referenced at first mention of the mutant proteins. Additionally, we have also now added a thermal stability assay which demonstrates that all of the RNF114 point mutants are stable, like the wildtype protein (Supplemental Figure 3G).

Minor

The mentioned analog sensitive chemical genetics is indicated as unpublished data (used for figure 1B). Will those data be included with the manuscript, or do the authors plan to publish this dataset at a later timepoint?

This dataset is part of a larger story focused on PARP7 biology, which is now available on BioRxiv (DOI 10.1101/2025.06.30.662483).

The authors discuss the K11-linked ubiquitin in Figure 1D, but it wasn't formally shown that this indeed is K11-linked. This can probably be confirmed by a K11-linkage antibody, DUB treatment or otherwise differently phrased - in later figures the authors do test the nature of linkages using DUBs.

We have previously shown that PARP7 is modified with K11-linked MARUbylation (Bejan et al 2025, EMBO J). We have revised the text to make this connection clearer.

The authors should perhaps reconsider their suggested naming "MUBD" and "MUTL". The DNA repair protein "MutL" exists in E.coli and using the same abbreviation might lead to confusion, even though the human version is named "MutL protein homolog 1". MUBD is not used as abbreviation yet as far as I know, but may be hard to pronounce when presenting these data?

We thank the reviewer for this information. The ubiquitin field has long accepted the abbreviation UBD for Ubiquitin Binding Domain; our MUBD nomenclature adds the mono-ADP ribose (M) and would be pronounced M-UBD. We have clarified this point in the text and better introduced UBD prior to our nomenclature. To better convey this new terminology, and to distinguish MUTL from MutL, we have adapted our abbreviated nomenclature to M-UBD and M-UTL throughout the revised manuscript.

Figure 4B/C, the figure legend is not quite clear. "After 30 minutes, the plate was removed from the instrument and samples were quenched in SDS sample buffer". Does this mean that the sample visualised in Figure 4C was taken at t=120? Also, in the text the authors describe that a {plus minus}50% reduction in signal was observed, but it is not clear how this value was calculated?

We apologize for the confusion. We have rewritten the sentence to say "30 minutes after DUB addition..." (t=90 minutes).

It would be good to indicate amounts of enzymes used in the figure legends. It is for example

indicated in the text that Cezanne was used at 20nM in some experiments, but I'd recommend to include this type of info with the figures as well.

This is a fair point. All other DUBs were used at 0.5 μ M and Cezanne was used at 20 nM in all experiments presented except the optimization shown in Supplemental Figure 4B. We have added this information to the figure legends.

The authors might want to include a little paragraph on how RNF166 functions in cells, considering that they do not see any effects of RNF166 knockdown in 1D? Might be worthwhile to expand the discussion to include some information about RNF166.

As addressed in a previous reviewer comment, we have used RT-qPCR to confirm RNF166 was knocked down to a similar level as RNF114 and DTX2. We can therefore more confidently suggest that RNF114 is the major extender of PARP7 MARUbylation under these cellular conditions. This demonstrates that, despite their similarities, RNF114 and RNF166 are not functionally redundant. We now elaborate on this point in the revised Discussion.

Referee #2:

Reviewer comments

The authors describe Ubiquitin E3 ligases that recognize specifically recently discovered dual post-translational modification where Ubiquitin is attached to mono-ADP-ribosyl groups added to proteins by PARP enzymes. The topic of the study is very relevant and timely and there are multiple studies ongoing and available as preprints currently as the authors also point out in the manuscript. The manuscript is well structured and has a logical progression between the experiments. The claims are mostly supported by the data provided, but there are a few major concerns that should be addressed before publication.

We thank the reviewer for their comments and constructive feedback.

Major concerns

Already in the title the authors talk about a family of E3 ligases but in the end, it is just two enzymes, RNF114 and RNF166, that have the described activity. The title should reflect the actual data better in this respect and be more specific.

Given the emerging complexity of this multilayered modification, we find it helpful to establish some nomenclature that describes the stepwise regulation (i.e., PARPs, DELTEXs, M-UTLs). As future work continues to build upon this complexity, having a set of terms such as these will provide a clear framework. While we could not formally show M-UTL activity for RNF138 and RNF125, their structural similarity and (at least for RNF138) binding to ADPr suggest that further studies may illuminate M-UTL or some similar function. Still, classification of mechanistically distinct families among Ub regulators is common practice in the field (e.g., "RCR" E3 ligases for MYCBP2, or "RZ" E3 ligases for RNF213 and ZNFX1), so classifying RNF114 and RNF166 as M-UTLs is appropriate and justified.

While the figures are visually nice, the authors should help the reader and label the figures better and also point out things to look at in the figure in the main text to improve readability. As examples figure 1A supernatant bands should be labelled like in Figure 4C and MARUbe is not MARUbe in the models but just Ub (like defined in the figure legends).

We are thankful for the advice. As other reviewers have also mentioned MARUbe labeling, we have changed this to be Ub_A in accordance with its role as the acceptor Ub in the K11 polyUb assembly reaction. We have also added additional labels to the blots in Figure 1, and elsewhere throughout the figures.

In Figure 1A, GFP WB detection exhibits a single band, as opposed to the Ub and MAR detection from which a smear is seen. Why is this? One could expect the signal to be low or even undetectable but not change behaviour. It is only in the input panel of Figure 1A where a smear is noticeable also in with the GFP antibody. It would be good to provide the full blots in the supplement.

Under the homeostatic cellular conditions tested, we suspect that only a small fraction of total GFP-PARP7 is MARUbylated. This explains why, at standard exposure levels, a GFP blot does not show the minor population that would smear upward with MARUbylation. This figure shows the entire blot of higher molecular weight than PARP7. We have provided the images in the source data.

In the context of Figure 1D, the authors provide verification of the knockdown in the supplementary information. This is, however, not convincing. It appears that DTX2 antibody is not working at all specifically (full blot not provided) and that siRNF166 is not functional. This puts into question the conclusions made that RNF114 would be the main E3 ligase as the RNF166 is always present in the experiments. The nonfunctional knockdown agrees well with the data presented. The claim that the blot had a high background is not justified in these cases that are some of the key results of the manuscript.

This is a great point also mentioned by other reviewers. We have since been able to confirm DTX2 knockdown by western blot using a different antibody. In lieu of a suitable antibody for RNF166, we demonstrate similar levels of knockdown for RNF166, DTX2, and RNF114 by RT-qPCR (Supplemental Figure 1). This validation of the siRNA knockdown allows us to assert more definitively that RNF114, and not RNF166, extends K11-linked MARUbylation onto PARP7 under our cellular conditions.

The binding of RNF138 to Ub-ADPrT is not convincing. Taken that RNFs are the about the same MW one would expect the anisotropy signal to be at the same level in Figure 5E. The observed binding appears to be unspecific. This is also in line with Figure 5F that shows RNF138 being inactive.

We respectfully disagree that the binding of RNF138 to Ub-ADPr-T is not convincing. This binding is weaker than RNF114, but there is still a preference of RNF138 in recognizing the Ub-ADPr over Ub. While the size of the protein complexes would be similar, the bulk FP signal is

also an indicator of the proportion of fluorescent protein bound into the complex. Thus, the smaller change in FP in this case simply reflects a weaker affinity.

Minor points and corrections

The language is good, but one could consider some adjectives to be removed to make it a bit more scientific (e.g. "strikingly", "newly").

We have edited our text to reflect this suggestion.

Please use TNKS1 and TNKS2 as acronyms for tankyrases and if the point is to mention that these belong to the same family add "formerly called PARP5a/b"

We have changed our nomenclature to reflect this suggestion.

In the Figure 1D the HA-Ub identification for the siDTX2 treatment seems to be in agreement with the claim that DTX2 plays a major role in the formation of the initial MARUbe as the 10 kDa band is mostly gone, but one would expect that this would also cause that the 20-25 kDa bands would have also had a considerable lower signal as there would be no place for them to dock. This is not the case, and the signal is similar to siRNF114 treatment. Is this related to another ligase that would be compensating or nonfunctional knockdown?

We have now more carefully quantified the monoUb-ADPr and polyUb-ADPr signals across 3 biological replicates and found a statistically significant reduction in all forms of MARUbe following DTX2 knockdown, as expected. Knockdown of RNF114, on the other hand, only reduces levels of polyUb-ADPr. In both cases we observe only a partial reduction, which could be a result of incomplete siRNA knockdown or functional redundancy among DELTEXs and/or M-UTLs. We elaborate on this point further in the revised text.

The last experimental section is a nice combination of evolutionary conservation and biochemistry to elucidate the activity in vitro of two of the newly described enzymes. However, given that in the discussion the poor conservation of RNF125 is mentioned again, would it have been possible to make a reconstruction of the evolutionary divergence of the enzymes? This would shed light on whether RNF125 is "evolving towards a different substrate" or if on the other hand, MARUbylation is a more recently developed system.

This is a great suggestion. We have looked at the evolutionary divergence of the four human enzymes and have included Figure 5D that demonstrates the divergence of RNF125.

Page 5, paragraph 1, line 7: remove phrases such as "a bit more complicated" for economy of language and reduce overall ambiguity.

We have made this change.

Page 5, paragraph 2, it would help to clarify in the text which lanes of the WB are being compared.

The text and figures have been modified to reflect this suggestion.

Figure 1B TssM cleavage site missing in the schematic.

This has been added.

Figure 1B, necessary to add the second Ub to make a visual clarification of the reason behind the two bands in the supernatant fractions?

This has also been added.

In Figure 1C please add the references to the studies to the figure legend. One cannot now evaluate one of the studies that is just referred to as "data unpublished". Would it be possible to add to be published elsewhere?

This dataset is part of a larger story focused on PARP7 biology, which is now available on BioRxiv (DOI 10.1101/2025.06.30.662483). We have added this reference to the text.

One should change "PARP7-specific inhibitor RBN2397" to "PARP7 inhibitor RBN2397" based on the supplementary information of the original publication.

We have made this change.

Page 7, paragraph 2, line 10-11, not necessary to direct the reader to MM?

We elected to direct readers to methods section in order to find additional detail on the production of Ub-ADPr, which is not a standardized process. Because we developed this strategy as part of the overall synthetic approach, we find it helpful to direct the readers in this way, rather than leaving it buried in the methods section.

Page 10, paragraph 1 (continuation of the last paragraph of Page 9), line 1, hereinafter would be more accurate.

We have made this change.

Page 14, paragraph 1 (continuation of last paragraph of page 13), line 7-8, refer to Perrard and Smith (2023) in which they also show that the UIM is necessary for the efficient modification of TNKS by RNF166.

We have included this reference.

Figure 5 B,C, the NAD and MARUbe (Ub as suggested in a previous point), could be re-arranged so as not to block the figure.

These labels have been adjusted.

Overall, one could change the yellow label in the MARUbe for a more visible colour.

We have darkened this label.

Having used structural predictions extensively, it would have been nice to see a full transfer complex with RNF138 that could perhaps indicate if indeed the linker length is crucial for the MUTLs to orientate ubiquitin correctly for the reaction to occur. One could also experiment by changing the linker length, but authors may want to reinspect the conservation surface also due to the lack of activity and that appears quite different from RNF166.

We indeed tried to model a full transfer complex for RNF125 and RNF138, but likely due to their distinct linker lengths, AlphaFold3 failed to model a high confidence model. We have added some information to the text to reflect these results.

It would be appropriate to add standard deviations for the measured dissociation constants.

As GraphPad Prism 10 does not provide standard deviations for the one-site binding fits, we have included the 95% confidence intervals containing the dissociation constants.

Referee #3:

The manuscript by Lacoursiere et al (from the laboratories of Michael Cohen and Jonathon Pruneda) focuses on deciphering how K11-linked ubiquitin chains are assembled on proteins that have been previously modified with ADP-ribose, that has then been ubiquitinated i.e. the target for K11 chain assembly already has an ADPr-Ub modification.

This study builds on prior work by these groups (Bejan et al., EMBO J 2025) and that of others. There is also a similar study available as a preprint <https://www.biorxiv.org/content/10.1101/2025.05.08.652854v1>

The key discovery described is the identification of RNF114 as an E3 ligase that binds ADP-ribose linked to Ubiquitin via two zinc binding domains and the C-terminal UIM (ubiquitin interaction motif). The development of labelled ADPr-Ub is critical to the approach taken and this reagent will likely enable other ADPr-Ub binding proteins to be characterised.

Having identified RNF114, Alphafold is used to prepare models of the complexes that are then supported by mutagenesis. Following binding to ADPr-Ub, RNF114 appears to then specifically promote the attachment of another ubiquitin molecule to Lys11 of the ADPr-Ub moiety. This is a remarkable example of modifying enzymes building considerable post-translational complexity on proteins. Following characterisation of RNF114 they then analyse other related E3 ligases and discover that RNF166 has similar properties to those of RNF114, but that the RNF138 and RNF125 - the other two related proteins differ.

The manuscript contains considerable data and will be of interest to those in the field, but as written it is not necessarily easily accessible to a wider audience. The modifications are complex and as a result care is required to allow figures to be readily interpreted - some general simple schematics might help. In addition, abbreviations should be reviewed - not all are helpful. In addition the authors may wish to consider the order of the text as in some cases reordering of paragraphs might more readily lead the reader through the figures.

We thank the reviewer for their comments and constructive feedback.

Main points:

1) The Abstract could be simplified to better articulate the key question addressed and the findings to make the research more readily available to a wider audience.

We have now revised the abstract to improve these qualities.

2) Figure 1A is complex and not well described. Close review of Bejan et al., EMBO J 2025 is required to interpret the figure. In addition, many parts of the figure are not described until later and the initial point made in paragraph 2 of the results takes some discerning for the non-specialist reader. It would be helpful to have another simple panel that introduces the reader to the modification before introducing the more complex experiment that follows from panel 1B. Alternatively the authors might consider reordering parts of the text and numbering the lanes so that specific lanes can be highlighted.

We thank the reviewer for highlighting these concerns. We have reworked Figure 1 and modified our schematic to describe the modifications and experiments.

3) Figure 1D is critical for the focus on RNF114, and while the data clearly suggests RNF114 is important, it does suggest that others E3s likely also have important roles as ~50% of modification remains following knockdown of RNF114. It is also interesting that the authors conclude RNF166 is not important for modification even though Supp Fig1 does not show evidence of knockdown - it seems an alternate conclusion could be that RNF166 was not reduced? The conclusions should be more carefully articulated with the caveats highlighted.

This point was raised by all reviewers. We have since modified this experiment and demonstrated a similar level of knockdown for RNF114, DTX2, and RNF166 by RT-qPCR, and have further validated that RNF114 and DTX2 protein levels were reduced by western blot. We still observe only a partial reduction in Ub-ADPr levels, which could be a result of incomplete siRNA knockdown or functional redundancy among DELTEXs and/or M-UTLs. We elaborate on this point further in the revised text.

4) The Alphafold models of the predicted complex with ADPr-Ub are compelling and the associated mutagenesis data is nice. The authors note that while F206 is predicted to make a key contact, the F206E mutation has little impact on banding. In the model F201 appear to be located nearby - is it possible that F201 occupies the position where F206 is modelled?

We agree that F201 also occupies the area around F206. We chose to focus on F206 since it lies in closer proximity to other important Ub residues including K11 and E34. Further, F206 is conserved in each ligase we studied here, whereas F201 is a leucine in RNF125.

5) In Figure 4 the E3 ligase activity is investigated and a model of the complex that accounts for the K11-linkage specificity proposed. This appears to nicely account for the assembly of K11-linked diUB but it is not clear how a K11-linked chain would be specified. The authors imply that longer chains can be built that have K11 links so it would be good to account for this or indicate uncertainty if the assembly of long chains is difficult to explain.

It is possible that after the formation of short K11-linked chains (as we observed to be the primary species in cells) that dissociation of the MARUbe permits the binding of the RNF114 UIM to an internal Ub in a chain. We have added a statement in the text to reflect this.

6) The data presented in Figure 4C is not as compelling as the related text. It might be helpful to highlight where the differences are and better explain the limitations of the data.

We have altered the text to more accurately describe these results. Although the polyUb isn't fully cleaved under these reaction conditions, Cezanne is the only DUB to show appreciable cleavage and therefore the conclusion remains.

7) Throughout Di19 is used to refer to the commonly described ZnF2 and ZnF3 domains in RNF114, and there is a proposal to name Di19 plus the UIM as MUBD. The term Di19 is not widely used and is somewhat a naming anomaly, and it is not clear that the term MUBD would bring clarity to the field. Furthermore the authors indicate that RNF125 is unlikely to bind ADPr-Ub, and RNF138 does not ubiquitylate Ub-ADPr, yet they propose to name the family of four proteins that includes RNF125 and RNF138 as MARUbe-targeted ligases (MUTLs). It is not clear that this is a helpful proposal.

Given the emerging complexity of this multilayered modification, we find it helpful to establish some nomenclature that describes the stepwise regulation (i.e., PARPs, DELTEXs, M-UTLs). As future work continues to build upon this complexity, having a set of terms such as these will provide a clear framework. While we could not formally show M-UTL activity for RNF138 and RNF125, their structural similarity and (at least for RNF138) binding to ADPr suggest that further studies may illuminate M-UTL or some similar function. Still, classification of mechanistically distinct families among Ub regulators is common practice in the field (e.g., "RCR" E3 ligases for MYCBP2, or "RZ" E3 ligases for RNF213 and ZNFX1), so classifying RNF114 and RNF166 as M-UTLs is appropriate and justified.

Minor points:

1) The first paragraph of the results is quite confusing and in part refers to the previous paper, but uses the phrase 'first we sought to'.

We have revised this region of the text to improve the clarity.

2) Figure 2 describes preparation of labelled ADPr-Ub that is a key tool that underpins

subsequent experiments. It might be helpful to number the steps in Figure 2A and refer to each in turn in the text. This would help the reader.

This is a great suggestion, and we have incorporated numbering to the text and schematic to guide readers.

3) It would be helpful to include a schematic of RNF114 with the location of all the mutations highlighted.

We have added this to Supplemental Figure 3.

4) In Figure 4A the preferential modification of ADPr-Ub is shown and the authors conclude that the ADPr-Ub linkage is recognised - it is also possible that it simply binds the linked substrate more tightly and not the linkage is recognised. This could be made clearer.

We have modified the text to reflect this suggestion.

5) The scale used for the PAE plot should be indicated.

We have added the appropriate scale bars to all PAE plots.

6) Throughout, but particularly in Figure 5 it would be helpful to have more whitespace between the panels in the Figure.

We have incorporated these changes.

Dr. Jonathan N Pruneda
Oregon Health & Science University
Molecular Microbiology & Immunology
3181 SW Sam Jackson Park Rd
Mail Code L220
Portland, OR 97239

1st Sep 2025

Re: EMBOJ-2025-121383R
A family of E3 ligases extend K11 polyubiquitin on sites of MARUbylation

Dear Jonathan and Mike,

Thank you for submitting your revised manuscript to The EMBO Journal. The three original referees have now assessed it once again (see comments below), and I am happy to say that they generally appreciate your revisions and responses. Referees 2 and 3 still retain a few specific concerns, which I feel should not require additional experimentation at this stage, but which would nevertheless warrant responding through a final point-by-point letter and presentational changes to text and figures, as appropriate. Especially regarding reservations about nomenclature and title, I'd be open to briefly discuss with you what might be the best options to satisfy these.

Apart from the referees' points, there are also a few editorial issues remaining to be addressed:

- Please carefully go through the reference list and make sure that each reference is complete with citation year, volume, and page/locator numbers - these are currently missing for several of them. Also, please make sure to fully adhere to EMBO Journal reference format (as stipulated in our Guide to Authors) throughout - for articles with more than 10 authors, only the first 10 should be listed, all others (even the last author) abbreviated with 'et al'.
- Please remove the header for Table EV1 from the legends section, it should only be contained in the Table file itself.
- Finally, please provide suggestions for a short 'blurb' text prefacing and summing up the study in two sentences (max. 250 characters), followed by 3-5 one-sentence 'bullet points' with brief factual statements of key results of the paper; they will form the basis of an editor-written 'Synopsis' accompanying the online version of the article. Please also upload a synopsis image, which can be used as a "visual title" for the synopsis section of your paper. The image should be in PNG or JPG format with the modest dimensions of EXACTLY 550 pixels wide and 300-600 pixels high.

I am therefore returning the manuscript to you once more for a final round of revision, asking you to upload the final modified files as quickly as possible, ideally by early next week, so that we might be able expedite acceptance and production of the study as much as possible.

With kind regards,

Hartmut

- 1) Every manuscript requires a Data Availability section (even if only stating that no deposited datasets are included). Primary datasets or computer code produced in the current study have to be deposited in appropriate public repositories prior to resubmission, and reviewer access details provided in case that public access is not yet allowed. Further information: embopress.org/page/journal/14602075/authorguide#dataavailability
- 2) Each figure legend must specify
 - size of the scale bars that are mandatory for all micrograph panels

- the statistical test used to generate error bars and P-values
- the type error bars (e.g., S.E.M., S.D.)
- the number (n) and nature (biological or technical replicate) of independent experiments underlying each data point
- Figures may not include error bars for experiments with $n < 3$; scatter plots showing individual data points should be used instead.

9) To facilitate reproducibility and cross-laboratory adoption of methodologies, please structure the Materials & Methods section as outlined in our guide to authors, including a completed Reagents and Tools Table that can be downloaded from our author guidelines as well (<https://www.embopress.org/page/journal/14602075/authorguide#structuredmethods>).

10) Digital image enhancement is acceptable practice, as long as it accurately represents the original data and conforms to community standards. If a figure has been subjected to significant electronic manipulation, this must be clearly noted in the figure legend and/or the 'Materials and Methods' section. The editors reserve the right to request original versions of figures and the original images that were used to assemble the figure. Finally, we generally encourage uploading of numerical as well as gel/blot image source data; for details see: embopress.org/page/journal/14602075/authorguide#sourcedata

In the interest of ensuring the conceptual advance provided by the work, we recommend submitting a revision within 3 months (30th Nov 2025). Please discuss the revision progress ahead of this time with the editor if you require more time to complete the revisions. Use the link below to submit your revision:

Link Not Available

Referee #1:

The authors have responded to all comments in an adequate manner, making this study a nice addition to the growing body of literature on "MARUbylation". I have no further comments or suggestions.

Referee #2:

Reviewer comments

The authors have considered the comments made by the reviewers and improved the manuscript. There is still, however, points that require further consideration before the manuscript can be published.

The authors justify well the description of the RNF114 and RNF166 as M-UTLs, but it should be made clear that these enzymes contain two domains Di19 + UIM and together they do not form a domain, but they are still two domains. Hence M-UBD acronym is not actually correct. At least the "domain" should be avoided in the name if it is really seen necessary to invent a new abbreviation. Calling this a "module" like in the abstract (M-UBM) would be more appropriate, but authors should also consider that there is an already defined ubiquitin binding motif (UBM).

The authors disagree with the point that RNF138 would not specifically bind to Ub-ADPr but the affinity would be higher. However, in figure 5 panel F the affinity of Ub-ADPr would be at max similar level to the RNF166 to Ub-Lys. Does that mean that Di19 of RNF138 is not functional? In any case stronger evidence of binding instead of what looks like unspecific binding curve, proper statistics and repetitions or alternative measurement techniques and measured affinities would be needed for this point of the authors to be valid.

Authors show the evolutionary divergence tree for the human versions of RNF114, RNF125, RNF138, and RNF166, but the point was more on the appearance of these proteins throughout evolution.

The authors decided to show a large confidence interval to the K_d measurements indicating that these measurements have not been repeated. It would be more appropriate to repeat the measurements three times and show the standard deviation of the values as now there is only one curve where the CI is only derived from the fit of that measurement.

Please add the positive charge to NAD⁺ labels and consider moving NAD from the top of the protein in Figure 5 panel B. There would be room next to NAD⁺ and it would not have a white block in an otherwise very nice figure.

Referee #3:

The authors have addressed most of my points and the manuscript is much improved.

One outstanding issue is the nomenclatures of various families/proteins/domains.

Another referee raised the issue of the title, which I agree is misleading and should be revised. There are many options but the data supports either 'RNF114 extends K11...' or 'RNF114 and RNF166 extend K11'.

All referees raised concerns about the nomenclature used in the manuscript. To address this the authors have strengthened their approach rather than modifying the approach taken.

Acronyms are a necessary part of biology but they can also cause confusion if care is not taken. I therefore remain unconvinced that renaming the family of proteins as MARUbe-targeted ligases (M-UTLs) is helpful when RNF125 is unlikely to bind ADPr-Ub, and RNF138 does not ubiquitylate Ub-ADPr.

The use of M-UBD to describe ZF2/ZF3/UIM is less problematic, although not necessarily helpful given the related manuscript in Kolvenbach et al., (Nature Chemical Biology, 2025) uses the term ZUD for the same region, while in Kloet et al. (Nature Comms 2025) they use ZnF2+ZnF3+UIM. At some point there will need to be agreement about the name. Given ZUD has already been used to refer to the zfDi19 and UIM maybe this would make more sense?

Lastly, the panel layout for some figures could still be improved by inclusion of more white space.

Referee #1:

The authors have responded to all comments in an adequate manner, making this study a nice addition to the growing body of literature on "MARUbylation". I have no further comments or suggestions.

We thank the reviewer for their constructive feedback that helped improve our manuscript.

Referee #2:

Reviewer comments

The authors have considered the comments made by the reviewers and improved the manuscript. There is still, however, points that require further consideration before the manuscript can be published.

The authors justify well the description of the RNF114 and RNF166 as M-UTLs, but it should be made clear that these enzymes contain two domains Di19 + UIM and together they do not form a domain, but they are still two domains. Hence M-UBD acronym is not actually correct. At least the "domain" should be avoided in the name if it is really seen necessary to invent a new abbreviation. Calling this a "module" like in the abstract (M-UBM) would be more appropriate, but authors should also consider that there is an already defined ubiquitin binding motif (UBM).

As this area of study is rapidly expanding, we find the definition of new terminology to be very helpful. While we respect the reviewer's stance that one definition of a "domain" is a structural unit, another definition is based instead on function. Because our work clearly demonstrates that the Di19 and UIM function as one unit, we find it most appropriate to consider them together. It is entirely possible that future work may identify a different combination of domains that confer binding specificity for MARUbe, and if that is the case, then those may still fall into a broad category of "M-UBD", similar to how UBD provides an encompassing term for many types of Ub-binding domain (e.g., UIM, MIU, UBA, etc.).

The authors disagree with the point that RNF138 would not specifically bind to Ub-ADPr but the affinity would be higher. However, in figure 5 panel F the affinity of Ub-ADPr would be at max similar level to the RNF166 to Ub-Lys. Does that mean that Di19 of RNF138 is not functional? In any case stronger evidence of binding instead of what looks like unspecific binding curve, proper statistics and repetitions or alternative measurement techniques and measured affinities would be needed for this point of the authors to be valid.

Although RNF166 and RNF138 are relatively similar in size, it is still not a fair comparison to cross-examine changes in fluorescence polarization across the two experiments. The longer linker region in RNF138, along with other sequence differences, could easily increase the conformational heterogeneity and thus decrease effective change in

polarization of the ligand-bound state. Combined with what is most likely a much lower overall affinity for ADPr, Ub, and Ub-ADPr, this likely contributes to the relatively weak changes in polarization. Even so, comparing within the RNF138 experiments shows a clear specificity for binding to Ub-ADPr over Ub or ADPr alone, which supports our limited conclusions that RNF138 contains a weak M-UBD and is a candidate M-UTL. We have added a statement to the results section that addresses the low changes in fluorescence polarization observed in our binding experiment.

Authors show the evolutionary divergence tree for the human versions of RNF114, RNF125, RNF138, and RNF166, but the point was more on the appearance of these proteins throughout evolution.

Apologies for the confusion. This is an interesting question. We have now investigated where in evolution these ligases appeared and included a short description in the Discussion section.

The authors decided to show a large confidence interval to the K_d measurements indicating that these measurements have not been repeated. It would be more appropriate to repeat the measurements three times and show the standard deviation of the values as now there is only one curve where the CI is only derived from the fit of that measurement.

The binding measurements were all repeated, as indicated by the data points shown alongside the binding curves. All replicates were used to calculate one non-linear regression, which we find to be a more robust approach. The large confidence intervals are due to incomplete saturation of binding, which was a limitation of our approach for some of the combinations with weaker affinities.

Please add the positive charge to NAD⁺ labels and consider moving NAD from the top of the protein in Figure 5 panel B. There would be room next to NAD⁺ and it would not have a white block in an otherwise very nice figure.

We have added the positive charge to the NAD⁺ labels, as suggested, and repositioned the labels in Figure 5.

Referee #3:

The authors have addressed most of my points and the manuscript is much improved.

One outstanding issue is the nomenclatures of various families/proteins/domains. Another referee raised the issue of the title, which I agree is misleading and should be revised. There are many options but the data supports either 'RNF114 extends K11...' or 'RNF114 and RNF166 extend K11'.

We appreciate the importance of descriptive titles and have updated ours to specifically mention RNF114 and RNF166.

All referees raised concerns about the nomenclature used in the manuscript. To address this the authors have strengthened their approach rather than modifying the approach taken.

Acronyms are a necessary part of biology but they can also cause confusion if care is not taken. I therefore remain unconvinced that renaming the family of proteins as MARUbe-targeted ligases (M-UTLs) is helpful when RNF125 is unlikely to bind ADPr-Ub, and RNF138 does not ubiquitylate Ub-ADPr.

The use of M-UBD to describe ZF2/ZF3/UIM is less problematic, although not necessarily helpful given the related manuscript in Kolvenbach et al., (Nature Chemical Biology, 2025) uses the term ZUD for the same region, while in Kloet et al. (Nature Comms 2025) they use ZnF2+ZnF3+UIM. At some point there will need to be agreement about the name. Given ZUD has already been used to refer to the zfDi19 and UIM maybe this would make more sense?

As discussed above, this area of study is rapidly emerging and careful definition of terms will be important for future work. We find it useful to refer to the mechanistically related binding domains and ligases that are part of this system. And while other names have appeared in the literature for the M-UBD, we find them to be less descriptive.

Lastly, the panel layout for some figures could still be improved by inclusion of more white space.

We appreciate this point, and have tried to balance this with maintaining a suitable size of our structure images.